# Accelerating Parallel Diffusion Model Serving with Residual Compression

**Jiajun Luo**[*]
Shenzhen International Graduate
School Tsinghua University
luo-jj24@mails.tsinghua.edu.cn

**Yicheng Xiao**[*]
Southern University of
Science and Technology
xiaoyc2022@mail.sustech.edu.cn

**Jianru Xu**
Southern University of
Science and Technology
12211830@mail.sustech.edu.cn

**Yangxiu You**
Jiangnan University
joesephyou@tencent.com

**Rongwei Lu**
Shenzhen International Graduate School
Tsinghua University
lurw24@mails.tsinghua.edu.cn

**Chen Tang**
The Chinese University of Hong Kong
chentang@link.cuhk.edu.hk

**Jingyan Jiang**
Shenzhen Technology University
jiangjingyan@sztu.edu.cn

**Zhi Wang**[†]
Shenzhen International Graduate School
Tsinghua University
wangzhi@sz.tsinghua.edu.cn

**Original**
Single-device

Latency: 23.16s

**CompactFusion**
Compression-based
*fresh activations*
FID = **3.263**
Latency: **7.57s**

**PipeFusion**
Overlap-based
*stale activations*
FID = 6.722
Latency: 9.49s

**DistriFusion**
Overlap-based
*stale activations*
FID = 9.911
Latency: 8.05s

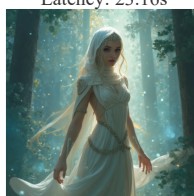 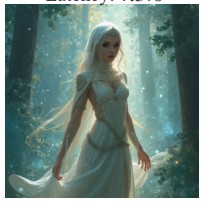 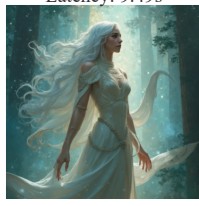 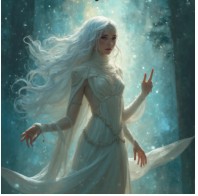

*Prompt: Ethereal fantasy concept art of an elf, magnificent, celestial, ethereal, painterly, epic, majestic, magical, fantasy art, cover art, dreamy.*

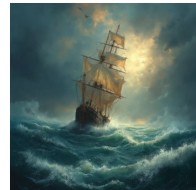 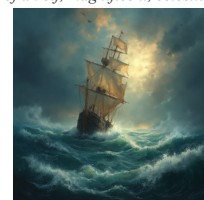 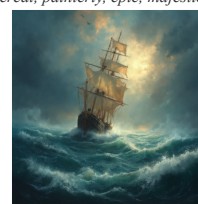 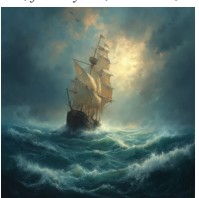

*Prompt: Romantic painting of a ship sailing in a stormy sea, with dramatic lighting and powerful waves.*

Figure 1: **CompactFusion** is a compression framework for parallel diffusion serving acceleration. Leveraging the intrinsic redundancy in diffusion models, CompactFusion transmits compressed step-wise residuals instead of full activation to significantly reduce communication volume and preserve fidelity, while prior works rely on stale activations for computation-communication overlap, leading to noticeable degradation. Setting: 4×L20, FLUX-1.dev, 28-step, 1024×1024 resolution, 1-step warmup for all algorithms.

[*]Equal contribution.
[†]Corresponding author.

39th Conference on Neural Information Processing Systems (NeurIPS 2025).

## Abstract

Diffusion models produce realistic images and videos but require substantial computational resources, necessitating multi-accelerator parallelism for real-time deployment. However, parallel inference introduces significant communication overhead from exchanging large activations between devices, limiting efficiency and scalability. We present **CompactFusion**, a compression framework that significantly reduces communication while preserving generation quality. Our key observation is that diffusion activations exhibit strong temporal redundancy—adjacent steps produce highly similar activations, saturating bandwidth with near-duplicate data carrying little new information. To address this inefficiency, we seek a more compact representation that encodes only the essential information. CompactFusion achieves this via **Residual Compression** that transmits only compressed residuals (step-wise activation differences). Based on empirical analysis and theoretical justification, we show that it effectively removes redundant data, enabling substantial data reduction while maintaining high fidelity. We also integrate lightweight error feedback to prevent error accumulation. CompactFusion establishes a new paradigm for parallel diffusion inference, delivering lower latency and significantly higher generation quality than prior methods. On 4×L20, it achieves $3.0\times$ speedup while greatly improving fidelity. It also uniquely supports communication-heavy strategies like sequence parallelism on slow networks, achieving $6.7\times$ speedup over prior overlap-based method. CompactFusion applies broadly across diffusion models and parallel settings, and integrates easily without requiring pipeline rework. Portable implementation demonstrated on xDiT is publicly available at `https://github.com/Cobalt-27/CompactFusion`.

## 1 Introduction

Diffusion models are scaling rapidly both in size and in computation, outpacing the capacity of single accelerators. The model sizes have grown from 983M parameters in Stable Diffusion 1.5 [1] to more than 12B in FLUX.1 [2]. Meanwhile, diffusion models are significantly more computationally intensive than LLMs, with compute increasing faster than the model size [3]. As computational demand rises, single-GPU inference can no longer meet latency constraints, making multiaccelerator parallelism essential for practical deployment.

However, parallel paradigms introduce a new challenge: the increasing prominence of communication bottlenecks, stemming from the fact that interconnect bandwidth has not kept pace with the growth of the compute. From A100 to H100, FP16 FLOPS increases over $6\times$ (312T $\rightarrow$ 1,979T), while NVLink bandwidth grows only $1.5\times$ and PCIe bandwidth simply doubles [4, 5]. In FLUX.1, standard patch parallelism transmits around 60 GB of activations per image per GPU, consuming over $45\%$ inference time across 4×L20 with PCIe interconnects. As bandwidth lags behind, communication increasingly dominates inference cost, underscoring the need to reduce transmission overhead.

Previous works exploited the **Temporal Redundancy** [6] inherent in diffusion models, where adjacent inference steps produce highly similar activations, to mitigate the communication bottleneck and accelerate parallel inference. Methods like DistriFusion [7] and PipeFusion [8] adopt **Displaced Parallelism**, which reuses stale activations from previous steps to overlap communication with computation. Although this reduces visible latency, it introduces three fundamental limitations. **(1)** Displaced parallelism replaces current activations with outdated ones, leading to noticeable quality degradation. **(2)** Moreover, the core data volume remains unreduced, meaning communication-intensive strategies like sequence parallel [9] still transmit large activations. Consequently, their performance gains are fragile and collapse when the overlap window insufficiently masks communication costs. **(3)** It requires nontrivial rework of the model pipeline [10], which makes integration complex and limits generality across architectures.

We rethink how temporal redundancy should be exploited, and propose **Residual Compression** as a more fundamental solution. Instead of merely overlapping the transfer of redundant activations, we ask: **why transmit redundant data at all?** Diffusion activations change slowly over time with adjacent steps producing highly similar value; yet prior methods still transmit full activations, saturating interconnects with near-duplicate data carrying little new information. Our key observation

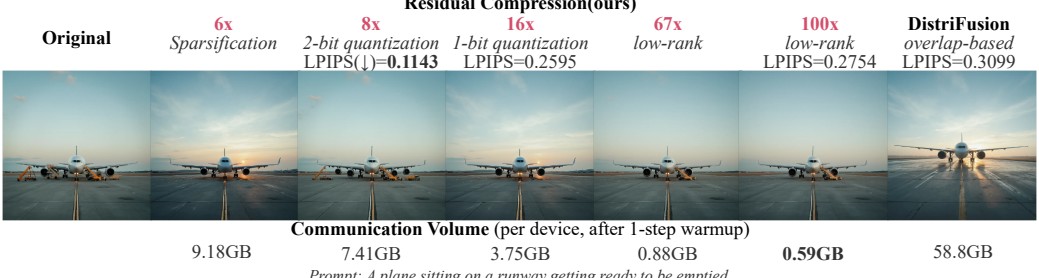

| | | Residual Compression(ours) | | | | DistriFusion |
|---|---|---|---|---|---|---|
| Original | 6x Sparsification | 8x 2-bit quantization LPIPS(↓)=**0.1143** | 1-bit quantization LPIPS=0.2595 | 67x low-rank | 100x low-rank LPIPS=0.2754 | overlap-based LPIPS=0.3099 |

**Communication Volume** (per device, after 1-step warmup)

| 9.18GB | 7.41GB | 3.75GB | 0.88GB | **0.59GB** | 58.8GB |
|---|---|---|---|---|---|

*Prompt: A plane sitting on a runway getting ready to be emptied.*

Figure 2: We introduce **Residual Compression** as a new paradigm for efficient parallel diffusion. It generalizes across compressors and sustains high quality even under $100.05\times$ compression. Setups: 4 devices, 1-step warmup.

is that removing redundant data should drastically reduce communication volume and resolve the communication bottleneck, while maintaining the quality. Thus, we seek a more compact representation that encodes only the essential change. Residual Compression achieves this by transmitting only compressed activation residuals (differences between time-steps), combined with a lightweight error feedback mechanism. Based on empirical analysis and theoretical justification, we show that it effectively removes redundant data, enabling substantial data reduction while maintaining high fidelity. We redefine how redundancy is addressed in parallel diffusion—not by hiding its transmission with overlap tricks, but by eliminating it at the source.

Residual compression surpasses previous overlap-based methods for following superiority:

- **Avoiding stale activations leads to higher quality.** Residual Compression works with current data, avoiding the quality degradation inherent in using stale activations in prior methods, yielding significantly higher fidelity. Under aggressive 2-bit quantization, residual compression achieves excellent quality , substantially outperforming state-of-the-art DistriFusion and PipeFusion in identical setups (Figure 1). Remarkably, even at extreme compression ratios of over $100\times$(transmitting $< 1\%$ of the original data), our method maintains better quality than DistriFusion (Figure 2 and section 4.2).

- **A significant drop in data volume yields much lower latency.** Residual Compression significantly reduces data exchange, delivering superior end-to-end performance across diverse hardware configurations. It achieved $3\times$ speedup on $4\times$H20 (NVLink) and $4\times$L20 (PCIe) clusters (Figure 1), surpassing prior methods. Furthermore, it makes communication-intensive parallel strategies practical even in low-bandwidth environments, achieving $6.7\times$ speedup over DistriFusion on Ethernet-level bandwidth.

- **Residual compression is structurally decoupled from the parallel pipeline, enabling broad compatibility and easy adoption.** It operates entirely at the communication layer, without modifying model logic or parallel execution flow. The design generalizes across compression methods (Figure 2) and parallel strategies, and has been applied on large-scale image and video models such as FLUX.1 and CogVideoX, and integrated into frameworks including xDiT and distrifuser, with fewer than 20 lines of core code changed. In contrast, prior methods are tightly coupled with specific strategies and require substantial rework. By focusing exclusively on the transmission aspect, residual compression maintains a lightweight profile and exhibits deployment readiness.

**CompactFusion** implements residual compression as a portable system, supporting compression techniques including quantization [11], low-rank, and sparsity (as shown in Figure 2), with tunable ratios from $2\times$ to over $100\times$ while preserving high fidelity. It works with parallel strategies such as Patch Parallel and Ring Attention, and integrates into existing frameworks (xDiT, distrifuser) with minimal modification. We believe CompactFusion represents a promising paradigm shift in parallel diffusion acceleration.

## 2   Preliminaries on Parallelism

Parallelism is the art of distributing tasks across multiple devices, enabling accelerators to work collaboratively for improved latency and throughput. This distribution can occur across various

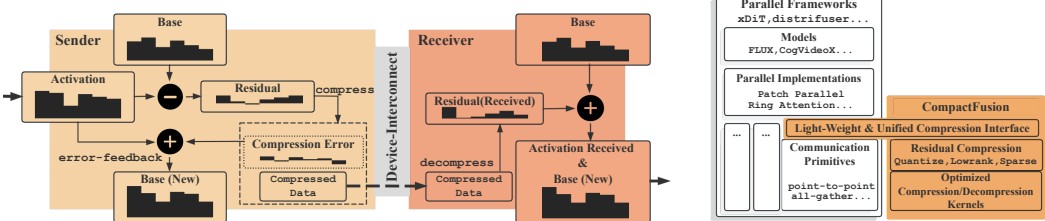

(a) Residual compression with error feedback.    (b) System overview.

Figure 3: (a) CompactFusion prevents long-term error accumulation by adding feedback into residual compression. (b) By wrapping a few standard communication primitives with a unified compression interface, CompactFusion requires minimal change to the existing frameworks and can easily be integrated to new model structure.

dimensions: input token(patch) sequence(Sequence Parallel [9]), model layers (Pipeline Parallel [12]), intra-layer computation (Tensor/Expert Parallel [13, 14]), or data samples (Data Parallel [15]). Notably, many parallel strategies necessitate intensive inter-device communication [7] due to the partitioning of weights, activations, or inputs among devices, requiring data exchange for complete computation.

This work focuses on **Sequence Parallel** [9, 16, 17], which partitions the input sequence across devices and is used in recent works on parallel diffusion [7, 8] as well as in high-performance diffusion inference frameworks [10, 18]. It is by far the **most widely adopted parallel strategy** for diffusion models, as it is highly latency-friendly and well suited for real-time generation. A more detailed description of these strategies is provided in Appendix B.

To reduce perceived latency, prior works overlaps communication with computation by reusing stale activations [7, 8, 19], but suffers from quality degradation, complex pipeline rework, and unchanged (potentially large) communication volume, limiting its effectiveness.

## 3 Method

CompactFusion mitigates the communication bottleneck in parallel diffusion by leveraging residual compression (Sections 3.1 to 3.3) and practical system co-design (Section 3.4), achieving significant data reduction with minimal quality degradation.

### 3.1 Exploiting Temporal Redundancy via Residual Compression

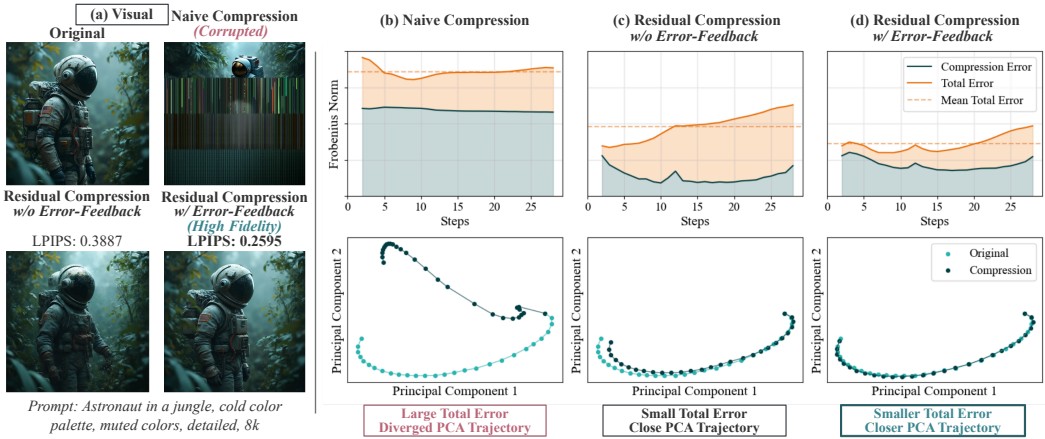

Figure 4: Comparison of strategies under 1-bit quantization: original (no compression), naive compression, residual compression with/without error feedback. Left: Visual Results, right: Error Analysis and PCA Trajectories (closer better) of activations over steps. We report two metrics: *Compression Error* (difference between input and output of compression) and *Total Error* (cumulative deviation from the uncompressed ground truth, lower is better). Setups: 4 devices, 1-step warmup.

Transmitting compressed residuals effectively eliminates redundancy from the data, lowering the communication volume while preserving fidelity. This efficiency stems from a key insight: diffusion models exhibit high temporal redundancy, with activations changing only minimally between steps. As a result, transmitting full activations often involves sending largely redundant information. By encoding only the meaningful differences, we can greatly reduce the volume of transmitted data. However, naively compressing high-dimensional activations still introduces substantial errors, often resulting in severe quality degradation or visual collapse (as shown in Figure 4(a)). Effectively leveraging temporal redundancy, therefore, requires more than simply reducing the amount of data—it demands meticulous design.

Our optimization builds on two observations. First, activations change gradually across steps, their step-wise residuals(differences from the previous result) having much smaller magnitude (as shown in Figure 5, residual norm is only a fraction of full activation norm). Second, compressing a smaller-magnitude signal typically induces proportionally less error. This makes the residual a more efficient compression target as it captures the essential change. We therefore propose residual compression: instead of transmitting the full activation, we transmit its compressed difference from the previous reconstruction and recover the current state by adding it back.

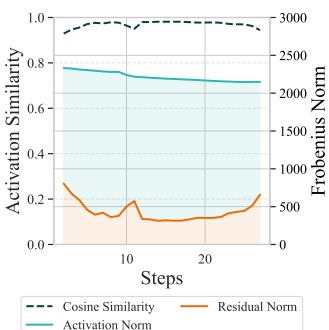

Figure 5: Adjacent-Step Activation Similarity and Activation/Residual Norm, on Flux.

To verify this principle, we apply aggressive 1-bit binarization (detailed in Appendix D) to both naive (compressing full activation, Figure 4(b)) and residual compression (Figure 4(c)) over Ring Attention. Naive compression fails completely: the output is corrupted, with large errors. In contrast, residual compression yields clean reconstructions with much smaller error at each step, and the activation trajectory remains close to the uncompressed baseline throughout.

While the idea of transmitting differences exists in other contexts [20, 21], to the best of our knowledge, CompactFusion is the first to demonstrate its effectiveness on parallel inference, and also the first on diffusion models, enabling aggressive data reduction while preserving generation quality.

**Limitation:** Despite low per-step error, residual compression accumulates error over time. As shown in Figure 4(c), the total error grows ovear steps without correction, which requires a mechanism to prevent the accumulation of long-term error.

## 3.2 Compensating Error Accumulation with Feedback

Residual compression significantly reduces per-step compression error by targeting smaller-magnitude signals. However, these small errors still accumulate over diffusion steps (curve for Total Error, Figure 4(c)). To eliminate long-term drift and preserve fidelity, CompactFusion integrates *Error Feedback*, a technique used in gradient compression [22, 23]. Rather than discarding the residual compression error at each step, we store it locally and add it to the next-step residual before compression, recirculating untransmitted information and systematically compensating for loss (illustrated in Figure 3a). This stabilizes reconstruction and prevents the compressed state from diverging from the original trajectory. Details are in Appendix H.

**Problem Formulation.** To understand how compression affects inference quality, we model diffusion as a sequence of transformations $a_t = f_t(a_{t-1})$, where $a_t$ denotes the intermediate activation in a fixed layer and $f_t$ captures the combined effect of one forward pass and scheduler update. Following relevant works [24, 25]. we define a $\delta$-compressor $C_\delta$ with $\delta \in (0, 1]$, satisfying $\mathbb{E}\left[\|C_\delta(z) - z\|^2\right] \leq (1 - \delta)\mathbb{E}\left[\|z\|^2\right]$ for $\delta \in (0, 1]$.

**Assumptions.** Our analysis relies on realistic assumptions grounded in empirical observations of diffusion behavior. First, the model exhibits strong local stability—captured by $L$-smoothness of each transformation $f_t$, i.e., $\|f_t(x) - f_t(y)\|^2 \leq L^2\|x - y\|^2$, with $L < 1$. Second, activations evolve gradually over time due to temporal redundancy: the expected per-step change is much smaller in scale than the activations themselves, i.e., $\sigma_\Delta^2 \ll \sigma_a^2$. We also assume uncorrelated error terms where appropriate. The definitions and assumptions are elaborated in Appendix H.2 and Appendix H.3;

these properties are well-supported in practice—activation norms remain bounded, and residual magnitude is significantly smaller than that of activations (Figure 5).

**Proposition 3.1** (Steady-State Error Bound). *Let $v^{naive}$ and $v^{residual}$ denote the steady-state mean squared error upper bounds under naive compression and residual compression with error feedback, respectively. Then under the given assumptions, their ratio satisfies the bound*

$$\frac{v^{residual}}{v^{naive}} = \frac{\sigma_\Delta^2}{\sigma_a^2} \cdot \frac{1 - L^2}{1 - L^2 - (1 - \delta)(L^2 + 1)}.$$

See Appendix H for the proof. This ratio confirms that residual compression with feedback yields a significantly lower steady-state error, often by an order of magnitude or more. The result holds under a mild stability condition: $\delta > 1 - \frac{1-L^2}{L^2+1}$ which imposes a lower bound on the required compression quality. In contrast, Residual compression *without* error feedback fails to converge; its error grows linearly with time steps and does not admit a steady state bound.

**Validation.** We validate this theory empirically using 1-bit binarization. As shown in Figure 4(d), CompactFusion with error feedback maintains low cumulative deviation throughout the inference process, closely follows the original trajectory, and generates high-quality images.

**Advantages.** Residual compression with error feedback enables robust and accurate compression under aggressive settings, maintaining fidelity even with 1-bit quantization ($16\times$ compression).

## 3.3 Scaling Residual Compression to Extreme Ratios

We aim to push the boundary of residual compression, targeting over $100\times$ compression ratio while preserving quality beyond state-of-the-art methods.

At extreme compression ratios, low-rank approximation is the only viable option. Quantization saturates at 1-bit ($16\times$), and sparsification collapses under diffusion—at $100\times$ sparsity, most values are never updated over typical 20–30 steps. In contrast, low-rank approximation provides a promising alternative: it retains full coverage of the coordinates while drastically reducing communication volume. Each activation or residual tensor $X \in \mathbb{R}^{n \times m}$ is approximated as $X \approx UV^T$, where $U \in \mathbb{R}^{n \times r}$ and $V \in \mathbb{R}^{m \times r}$, with $r \ll \min(n, m)$.

To make this feasible under real-time constraints, we adopt subspace iteration [26] (detailed in Appendix D), a faster alternative to SVD. While SVD offers optimality, it is approximately 60 times slower than the subspace iteration solution, far exceeding the $\sim$5ms communication window and $\sim$1ms compression budget in practice. In contrast, we adapt the subspace iteration from gradient compression [27], producing a usable approximation in milliseconds.

We improve low-rank compression by achieving a better tradeoff between precision and directional coverage. We observe that diffusion residuals are frequently high-rank, indicating that the activation changes span many directions across steps. However, each transmission is constrained to a low-rank subspace. We identify this high-rank/low-rank mismatch as a key bottleneck: the model attempts dense updates, but the compressor delivers sparse (in terms of subspace dimension) projections. To resolve this, we propose a strategic tradeoff—sacrificing accuracy (closeness to the optimal approximation) to expand rank coverage (number of dimensions covered per step). Concretely, we quantize the low-rank matrices with INT4, allowing higher per-step rank under the same transmission budget. This transforms the compressor into a subspace throughput machine, delivering broader coverage across update directions, even if each is less precise.

We empirically validate this strategy by comparing it with an alternative that improves approximation optimality through more subspace iterations(see Appendix F.1). Despite added quantization error, expanding rank coverage yields significantly better generation quality, confirming that directional breadth matters more than per-step accuracy under tight bandwidth constraints.

Notably, our approach maintains better visual fidelity over DistriFusion even at $100.05\times$ compression on FLUX.1-dev.

## 3.4 System Co-Design for Efficiency and Usability

CompactFusion is co-designed for real-world deployment—translating algorithmic gains into measurable latency reduction with minimal integration cost.

**Optimized Compression Kernels.**    We develop highly optimized GPU kernels for our compression techniques. Beyond efficient quantization and subspace iteration, we repurpose the concept of N:M sparsity [28] which originally used in weight pruning, and adapt it to activation compression via a N:M block sparsifier. Unlike standard TopK methods that suffer from sorting overhead and irregular memory access, our block sparsifier enables linear-time selection and GPU-friendly memory patterns. It is the only sparsifier design we found to yield practical on-device speedup.

**Latency Hiding via Overlap.**    The framework runs compression kernel concurrently with communication primitives. By carefully integrating into Ring Attention pipeline, it effectively overlap the compression overhead with communication waits, minimizing its impact on latency.

**Ease of Integration.**    CompactFusion benefits from the inherent modularity of Residual Compression, which cleanly separates compression from execution logic (Figure 3b). It wraps standard communication primitives without altering model code or parallel pipeline, and generalizes across strategies, compressors, and frameworks. In contrast, prior methods often entangle with specific parallel strategies and require substantial pipeline rework.

## 4    Experiments

### 4.1    Setups

**Models.**    Our method works with off-the-shelf models, we evaluate it on the state-of-the-art FLUX.1-dev [2] for image generation, and on CogVideoX-2b [29] for video generation. We adhere to standard inference settings on xDiT, employing 28 steps for FLUX.1-dev and 50 steps for CogVideoX-2b.

**Dataset.**    We test the image generation model using prompts in COCO Captions 2014 dataset [30] and video generation using prompts sampled from VBench [31].

**Hardware.**    To demonstrate broad applicability, experiments are conducted on various hardware and interconnects: high-bandwidth NVLink (H20 clusters, bandwidth: 366 GB/s), standard PCIe (L20 clusters, bandwidth: 17.13 GB/s) and simulated lower-bandwidth Ethernet (A40 clusters using `tc` for traffic control), ensuring robustness evaluation under various deployment constraints.

**Baselines.**    We apply CompactFusion to Sequence Parallelism, the most widely adopted strategy in parallel diffusion (Section 2), to reflect real-world deployment scenarios. We compare it with strong baselines from both industry and research, including **Patch Parallel** (all-gather KV) [18], **Ring Attention** (ring-style KV transfer) [9], **DeepSpeed-Ulysses** (all-to-all) [16], and **DistriFusion** (displaced patch parallel) [18]. We also include **PipeFusion** (displaced TeraPipe [32]) [8] as a baseline, though it is orthogonal to our method. We note that PipeFusion is not evaluated on video models as the xDiT does not support PipeFusion for video models (further discussed in Appendix C).

**CompactFusion Variants.**    We showcase three CompactFusion variants over Sequence Parallel with Ring Attention. **Compact-1bit** and **Compact-2bit** apply 1-bit and 2-bit quantization, with $16\times$ and $8\times$ compression respectively. **Compact-Lowrank** shows the potential for extreme communication reduction, reaching $100.05\times$ compression on FLUX via rank-32 approximation and INT4.

**Metrics.**    Following the relevant work, we adopt the standard evaluataion metrics for image quality testing: Peak Signal Noise Ratio (PSNR), Learned Perceptual Image Patch Similarity (LPIPS) [33] and Fréchet Inception Distance (FID) [34]. We adopt the following metrics for video quality testing: Structural Similarity Index Measure (SSIM)[35], Peak Signal-to-Noise Ratio (PSNR), and Learned Perceptual Image Patch Similarity (LPIPS)[33]. We also conduct a human evaluation study to assess perceptual consistency from a human perspective (detailed in Appendix E.1).

### 4.2    Main Results

CompactFusion achieves lower latency while maintaining high visual fidelity. It consistently performs well across different generation models, including FLUX.1-dev for images and CogVideoX for videos. It is also robust across hardware setups—L20, H20, and A40—and across network conditions such as

NVLink, PCIe, and simulated Ethernet. These results are shown in Figure 6. Quantitative metrics across 3, 4, and 6-GPU scales are provided in Tables 1 and 2.

Notably, CompactFusion uniquely enables the widely adopted but communication-intensive sequence parallel over slow networks, achieving up to $6.7\times$ speedup over DistriFusion under Ethernet-level bandwidth conditions (Figure 7).

By directly eliminating redundant data, CompactFusion drastically reduces the transmission volume. In its low-rank variant, it sends less than 1% of the original activation data. Despite this, it still outperforms DistriFusion in generation quality, as shown in Table 1 (image) and Table 2 (video).

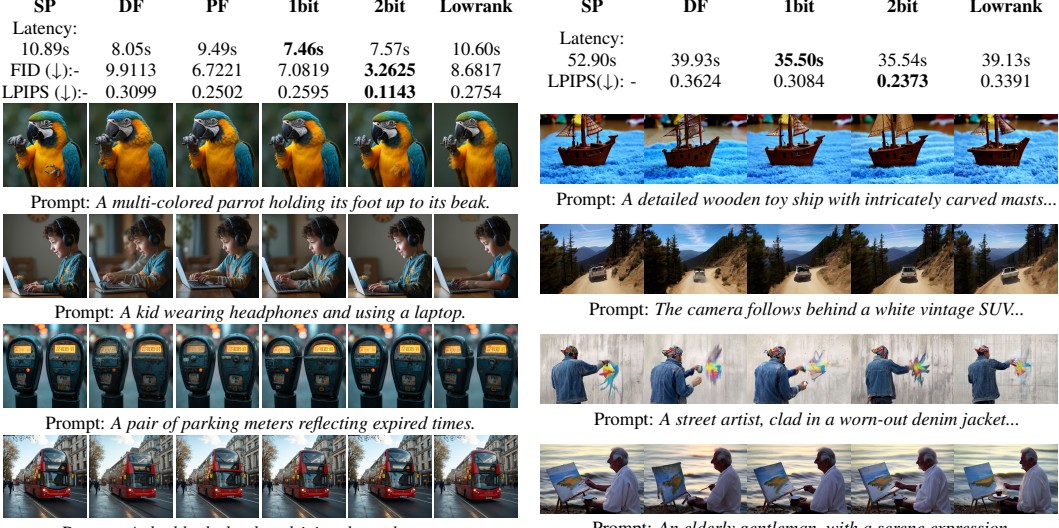

Figure 6: Qualitative results. FID and LPIPS is computed against the original images. **SP** stands for Sequence Parallel using Ring Attention, **DF** for DistriFusion and **PF** for PipeFusion. **1bit**, **2bit** and **Lowrank** are our methods. We use 1-step warmup for CompactFusion/PipeFusion/DistriFusion.

Table 1: Quantitative evaluation on 4 GPUs. **w/ G.T.** means calculating the metrics with the ground-truth COCO images, while **w/ Orig.** means with the original model's samples. Latency results on L20 (PCIE) and H20 (NVLink) are reported; speedup is calculated with respect to the single-device latency. Note that COCO-based metrics w/ G.T. may be uninformative. Its variations are small, and even disrupted outputs can score well [36].

| Method | PSNR (↑) | LPIPS (↓) | | FID (↓) | | Latency (s) | | Speedup | | Human |
|---|---|---|---|---|---|---|---|---|---|---|
| | w/ Orig. | w/ G.T. | w/ Orig. | w/ G.T. | w/ Orig. | L20 | H20 | L20 | H20 | Eval. (↑) |
| Original(Single-device) | | | | | | 23.16 | 20.26 | 1.00 | 1.00 | |
| Ring Attention | – | 0.772 | – | 32.75 | – | 10.89 | 7.54 | 2.12 | 2.68 | – |
| Patch Parallel | | | | | | 10.90 | 6.83 | 2.12 | 2.97 | |
| DeepSpeed-Ulysses | | | | | | 9.13 | **6.70** | 2.54 | **3.02** | |
| DistriFusion | 21.63 | 0.761 | 0.310 | 33.12 | 9.91 | 8.05 | 8.05 | 2.88 | 2.52 | 0.25 |
| PipeFusion | 23.42 | 0.766 | 0.250 | 32.36 | 6.72 | 9.49 | 9.07 | 2.44 | 2.23 | 0.56 |
| **Compact-1bit(Ours)** | 22.90 | 0.767 | 0.260 | 33.20 | 7.08 | **7.46** | 6.86 | **3.10** | 2.95 | 0.47 |
| **Compact-2bit(Ours)** | **29.54** | 0.772 | **0.114** | 33.09 | **3.26** | 7.57 | **6.70** | 3.06 | **3.02** | **0.84** |
| Compact-Lowrank(Ours) | 22.85 | 0.769 | 0.275 | 33.07 | 8.68 | 10.60 | 11.99 | 2.18 | 1.69 | 0.38 |

## 4.3 Ablation Studies

We perform targeted ablation studies to evaluate the contributions of core components and design choices in CompactFusion.

**Warmup Step.**   Like displaced parallel, CompactFusion requires at least a 1-step warmup, where the uncompressed activation is used to initialize the base tensor for later residual computation (detailed in Appendix D). Figure 8 compares different methods and warmups. We observe that CompactFusion maintains stable and high visual quality with just a single warmup step, showing

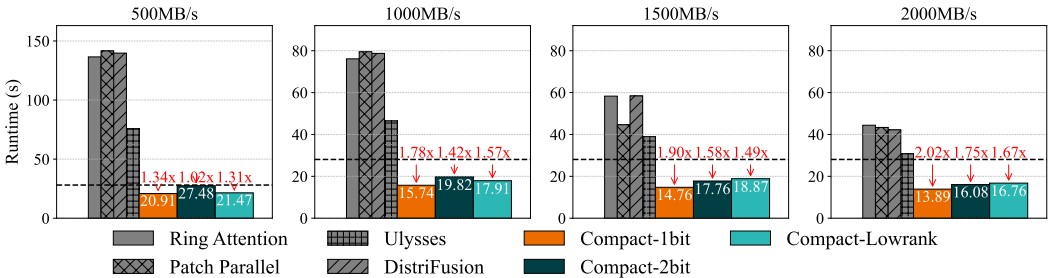

Figure 7: Measured latency of sequence parallel variants on 4×A40 GPUs under simulated Ethernet bandwidth . Horizontal dashed line indicates single-device latency. Only CompactFusion yields speedups, uniquely enabling efficient sequence parallelism over slow networks.

Table 2: Quantitative video generation evaluation on 3 GPUs and 6 GPUs. All metrics (SSIM, PSNR, LPIPS) are calculated with respect to the original samples. Latency results on L20 (PCIE) are reported; speedup is calculated against single-device latency.

| Device | Method | SSIM (↑) | PSNR (↑) | LPIPS (↓) | Latency (s) | Speedup |
|---|---|---|---|---|---|---|
| 1 | Original(Single-device) | – | – | – | 122.42 | – |
| 3 | Ring Attention | – | – | – | 63.30 | 1.93× |
| | Patch Parallel | – | – | – | 72.61 | 1.69× |
| | DeepSpeed-Ulysses | – | – | – | 62.65 | 1.95× |
| | DistriFusion | **0.842** | 24.13 | 0.220 | 60.75 | 2.01× |
| | **Compact-1bit (Ours)** | 0.762 | 20.46 | 0.291 | **56.41** | **2.17×** |
| | **Compact-2bit (Ours)** | 0.832 | **24.37** | **0.217** | 56.73 | 2.16× |
| | Compact-Lowrank (Ours) | 0.728 | 19.64 | 0.320 | 60.98 | 2.01× |
| 6 | Ring Attention | – | – | – | 52.90 | 2.31× |
| | Patch Parallel | – | – | – | 53.37 | 2.29× |
| | DeepSpeed-Ulysses | – | – | – | 39.03 | 3.14× |
| | DistriFusion | 0.703 | 18.09 | 0.362 | 39.93 | 3.07× |
| | **Compact-1bit (Ours)** | 0.746 | 19.75 | 0.308 | **35.50** | **3.45×** |
| | **Compact-2bit (Ours)** | **0.813** | **23.26** | **0.237** | 35.54 | 3.45× |
| | Compact-Lowrank (Ours) | 0.708 | 19.06 | 0.339 | 39.13 | 3.13× |

little degradation compared to longer warmup. In contrast, baseline methods are more sensitive to warmup configuration. For more detailed results, please refer to Appendix F.3

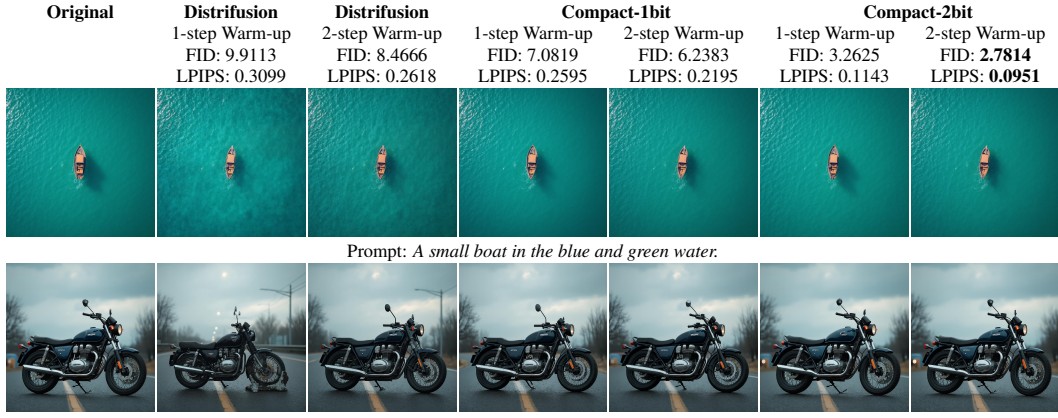

Prompt: *A small boat in the blue and green water.*

Prompt: *A motorcycle sits on the pavement on a cloudy day.*

Figure 8: Impact of warmup steps on visual quality.

**Effectiveness of Error Feedback.** Error Feedback significantly improves quality. For Compact-1bit under 1-step warmup, it reduces FID from 19.23 to 7.08, lowers LPIPS from 0.389 to 0.260, and increases PSNR from 19.78 to 22.90. For more detailed results, please refer to Appendix F.2.

**Performance on Patch Parallel and Ring Attention.** CompactFusion supports both Patch Parallel and Ring Attention with minimal integration effort. On FLUX.1-dev and 4×L20, Compact-2bit

reduces latency from 10.90s to 8.08s on Patch Parallel ($26\%$ reduction), and from 10.89s to 7.71s on Ring Attention ($29\%$ reduction). In both settings, it delivers high-quality generation, significantly outperforming prior work across perceptual metrics.

## 5   Conclusion

CompactFusion accelerates parallel diffusion by eliminating temporal redundancy at the source. It transmits only compressed residuals with error feedback, preserving generation quality while significantly reducing communication. CompactFusion is designed as a modular, drop-in layer over standard communication primitives, and integrates seamlessly into existing frameworks. Extensive experiments show that it delivers consistent speedups across models, hardware, and networks. We offer a lightweight and efficient approach for diffusion inference at scale.

## 6   Acknowledgements

This work is supported in part by the National Key Research and Development Project of China (Grant No. 2023YFF0905502), National Natural Science Foundation of China (Grant No. 92467204 and 62472249), and Shenzhen Science and Technology Program (Grant No. JCYJ20220818101014030 and KJZD20240903102300001). This study is also supported in part by the Natural Science Foundation of Top Talent of SZTU(Grant No. GDRC202413).

We would like to express our sincere appreciation to the xDiT team for their comprehensive parallel diffusion framework, which served as the foundation for our experiments.

We thank all participants who contributed to the human evaluation survey. We also thank Zhengyi Su for helpful discussions and support on the theoretical analysis.

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

# A  Limitations

While CompactFusion offers strong compression with minimal quality loss, several practical limitations remain.

First, compression itself introduces overhead. Although our implementation overlaps compression with communication in Ring Attention, full overlap is not always achievable, especially on fast interconnects with limited idle time. This can reduce net gains. We believe further kernel optimizations and scheduling strategies can help bridge this gap.

Second, while our low-rank variant supports high compression ratio, the subspace iteration it relies on remains slower than quantization or sparsity. This limits its speedup under fast networks, though still valuable for bandwidth-limited settings like edge deployment.

# B  More Preliminaries

## B.1  Sequence Parallelism

Sequence parallel enables multiple GPUs to collaboratively generate a single image or video sample, thereby reducing the latency per sample. During inference, an image or video frame is first divided into a sequence of patches (tokens). Sequence parallel partitions this sequence of patches across multiple devices so that each GPU is responsible for computing a section of it.

After the partition, however, inter-GPU communications are required. For FFN layers, communication is unnecessary—each patch (token) is processed independently, and all required data resides locally on the same device. But for attention layers, every patch must attend to all other patches across the full sequence, which introduces cross-device dependencies.

Different parallel strategies handle this communication differently:

**Patch Parallel[7].** This strategy performs a single `AllGather` operation over all devices to collect the complete set of keys and values (Figure 9). Each GPU thus receives global $K$ and $V$ tensors before executing the attention computation. It may incur substantial communication overhead as the entire $K, V$ matrices are transmitted every step.

**Ring Attention[9].** Instead of a full `AllGather`, Ring Attention organizes the devices into a logical ring and performs a series of peer-to-peer exchanges. Each stage transmits only a shard of $K, V$ and computes the corresponding partial result. Computation and communication are interleaved, offering partial overlap.

**Ulysses[16].** Ulysses adopts a slightly different design, using two `All-to-All` operations to exchange the required data between devices.

We provide illustrative pseudocode in Algorithms 1 and 2.

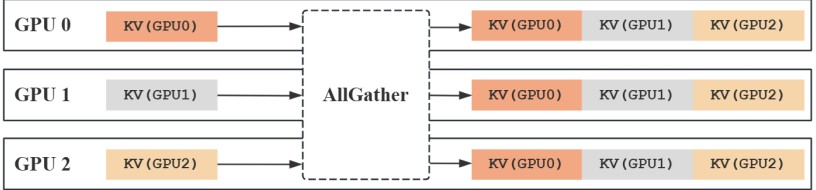

Figure 9: Illustration of Patch Parallel.

---

**Algorithm 1** Patch Parallel

---

**Input:** Local tokens/features $X_i$ on device $i$
1: $Q_i, K_i, V_i \leftarrow \text{ProjectQKV}(X_i)$
2: $K_{\text{gathered}}, V_{\text{gathered}} \leftarrow \texttt{AllGather}(K_i, V_i)$
3: $O_i \leftarrow \text{Attention}(Q_i, K_{\text{gathered}}, V_{\text{gathered}})$
4: **return** $O_i$

---

---

**Algorithm 2** Ring Attention

---

**Input:** Local tokens/features $X_i$ on device $i$, total devices $N$

1: $Q_i, K_i, V_i \leftarrow \text{ProjectQKV}(X_i)$
2: Initialize local output buffer: $O_i \leftarrow 0$
3: **for** $s = 0$ **to** $N{-}1$ **do**
4:     $K^{(s)}, V^{(s)} \leftarrow \text{RingSendRecv}(K_i, V_i, s)$        ▷ receive shard from device $(i{-}s) \bmod N$
5:     $O_i \leftarrow \text{Aggregate}\big(O_i, \text{AttentionPartial}(Q_i, K^{(s)}, V^{(s)})\big)$
6: **end for**
7: **return** $O_i$

---

## B.2 Displaced Patch Parallel (DistriFusion)

Patch Parallel performs a synchronous `AllGather` of $K, V$ at every denoising step and waits immediately before attention, which leads to substantial blocking when the transfer payload is large. Displaced Patch Parallel [7] avoids this by adopting a *send-now, receive-later* schedule with a one-step displacement. In each iteration, the system launches a non-blocking communication for the *current* $(K, V)$ but computes attention using the *previous* step's gathered $(K, V)$, thereby overlapping communication with computation. Each iteration waits on the handle from the earlier launch when those tensors are actually required. This schedule reduces latency at the cost of using stale keys and values, which can introduce quality drift.

## C Discussion

**Residual Compression and Displaced Parallelism.** Residual compression and displaced parallelism are not mutually exclusive. They can potentially be combined to leverage the strengths of both. For example, integrating CompactFusion with PipeFusion may allow reducing the memory footprint needed to store base tensors for residual computation.

**Exploiting Temporal Redundancy.** Inspired by excellent prior work DistriFusion and PipeFusion, we revisit the core idea behind temporal redundancy. As the name suggests, this data is largely redundant. Rather than continuing to transmit it and try masking its communication cost, we believe the most effective approach is to eliminate it entirely from the communication path. CompactFusion follows this principle: it directly discards redundant data, leading to more efficient and scalable parallel diffusion.

**Why PipeFusion is not evaluated on video models.** To the best of our knowledge, xDiT is the most widely used parallel diffusion inference framework, actively maintained and widely adopted in both research and deployment. It originates from the official PipeFusion repository and inherits its core features. However, neither PipeFusion nor xDiT currently support video models such as CogVideoX. As a result, evaluating PipeFusion in this setting would require substantial reengineering. This is beyond the scope of our work and the capacity of us. We stress that the omission is due to technical constraints rather than intentional bias. Our goal remains fair and comprehensive comparison wherever feasible.

## D Implementation Details

**Warmup Implementation.** Similar to displaced parallelism, CompactFusion requires at least one warmup step. It is used to initialize the uncompressed base tensor for residual calculation. Ablation studies show that CompactFusion is highly robust and achieves strong generation quality even with a single warmup step (Section 4.3). There are two practical ways to implement warmup: (1) running the warmup steps with parallelism but without compression; or (2) letting each device execute the warmup locally without any communication. The first method introduces some communication but is typically faster on PCIe and NVLink. However, it may become a bottleneck under extremely slow networks. The second method avoids communication but incurs slightly higher latency due to lack of parallelism. We adopt the first method in our experiments.

**Quantization.** We quantize an activation tensor $X \in \mathbb{R}^{N \times C}$ using element-wise multiplication between a quantized sign/value tensor and a scale matrix:

$$Q(X) = q(X) \odot (uv^\top),$$

where

- $q(X)$ is the element-wise quantized tensor (e.g., $\pm 1$ for 1-bit, or $\{-2, -0.5, +0.5, +2\}$ for 2-bit),
- $\odot$ denotes the element-wise product,
- $u \in \mathbb{R}^{N \times 1}$, $v \in \mathbb{R}^{C \times 1}$ form a rank-1 approximation of the activation magnitude.

In practice, we estimate $u$ and $v$ as:

$$u_i = \frac{\frac{1}{C} \sum_j |X_{ij}|}{\frac{1}{NC} \sum_{i,j} |X_{ij}|}, \qquad v_j = \frac{1}{N} \sum_i |X_{ij}|.$$

This yields a scale matrix where $(uv^\top)_{ij}$ reflects the product of the normalized token-wise and channel-wise means. All quantization operations are implemented in fused end-to-end Triton kernels for efficiency. We note that this scale estimation is heuristic and leaves substantial room for improvement.

**Low-Rank Compression.** We aim to approximate a matrix $A \in \mathbb{R}^{m \times n}$ with a low-rank decomposition of the form:

$$A \approx UV^\top, \quad \text{where } U \in \mathbb{R}^{m \times r}, \ V \in \mathbb{R}^{n \times r}.$$

To compute this approximation efficiently, we use a $T$-step subspace iteration [26]; subspace iteration was first used in gradient compression [27]:

---

**Algorithm 3** Subspace Iteration (rank-$r$)

---

**Input:** $A \in \mathbb{R}^{m \times n}$, target rank $r$, iterations $T$
1: Randomly Sample $Q$ and orthonormalize: $Q \leftarrow \text{orthogonalize}(Q)$.
2: **for** $t = 1$ **to** $T$ **do**
3: $\quad Z \leftarrow A^\top (A Q) \in \mathbb{R}^{n \times r}$
4: $\quad Q \leftarrow \text{orthogonalize}(Z)$
5: **end for**
6: $U \leftarrow A Q \in \mathbb{R}^{m \times r}$
7: $U \leftarrow \text{orthogonalize}(U)$
8: $V \leftarrow Q \in \mathbb{R}^{n \times r}$
9: **return** $U$, $V$

---

**Residual Base and Error Feedback.** In CompactFusion, each transmitted tensor represents a residual, which is the change in activation relative to a stored base. The base is initialized with the uncompressed activation during the warm-up step and updated after every iteration. Compression introduces an error that is observable only on the sender side, since the receiver never sees the true residual before compression. Therefore, error feedback is applied at the sender: the previously accumulated error is added back to the next residual before compression, ensuring that lost information is gradually compensated over time.

**Integration into Parallel Pipelines.** CompactFusion integrates into existing parallel frameworks by simply wrapping standard communication primitives such as `send/recv` and `AllGather`. This modular design allows the same compression interface to be reused across different parallel strategies, ensuring compatibility with existing diffusion inference pipelines.

## E   More Experiment Details

### E.1   Human Evaluation

To complement automated perceptual metrics (e.g., FID, LPIPS), we conducted a human evaluation study to assess how different parallel inference strategies affect generation quality.

We distributed an online questionnaire to colleagues and friends, asking them to compare images generated by five different parallelization methods. The goal was to identify which results were more visually consistent with the original reference image.

Directly ranking five images per question was found to be cognitively demanding and time-consuming, which discouraged participation. To simplify the task while still capturing ranking preference, we designed an alternative method. We randomly sampled 30 groups of images, and each participant answered 30 questions (15 of each type), with the following two instructions:

- Please select the two images that are *most* similar to the Reference Image.
- Please select the two images that are *least* similar to the Reference Image.

Each question displayed one reference image (generated by the original uncompressed model) and five test images produced by different parallel strategies (including DistriFusion, PipeFusion, and CompactFusion variants), as shown in Figure 10. The five options were randomly shuffled to avoid position bias. The prompts used for generation were randomly sampled from the COCO dataset.

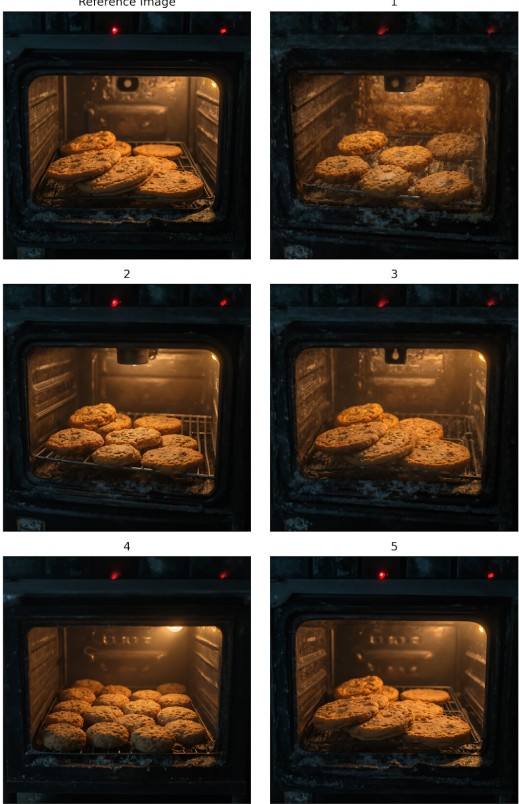

Figure 10: Example from our human evaluation questionnaire. Participants are asked to select two images most (or least) similar to the reference image. Each set contains five outputs generated by different parallel inference methods, shown in random order.

We collected 20 valid responses in total. For each method, we compute the probability that its output falls into the more consistent subset (either selected as most similar, or not selected as least similar). This reflects how frequently a method produces perceptually closer results to the reference image.

We note that this evaluation involved no personal or sensitive information and posed no potential risks to participants.

## E.2   More Setups

**Models.**   The default scheduler used is DPM solver. [37]

**Dataset.**   For evaluation, we randomly sample 5000 prompts from the image validation set and 200 video validation prompts.

# F   More Quantitative Results

This section provides more quantitative results for ablation study and some experiments we use to elaborate our methods.

## F.1   Effect of Tradeoff between Rank and Precision

Table 3: Quantitative evaluation for Compact-Lowrank, Pure Lowrank with Iteration 2 and Pure Lowrank with Iteration 10.

| Method | PSNR (↑) | | LPIPS (↓) | | FID (↓) | |
|---|---|---|---|---|---|---|
| | w/ G.T. | w/ Orig. | w/ G.T. | w/ Orig. | w/ G.T. | w/ Orig. |
| Compact-Lowrank | 9.89 | 22.85 | 0.769 | 0.275 | 33.07 | 8.68 |
| Pure Lowrank 12 with Iteration=2 | 9.86 | 20.78 | 0.765 | 0.340 | 33.91 | 12.19 |
| Pure Lowrank 8 with Iteration=2 | 9.84 | 20.11 | 0.762 | 0.361 | 34.68 | 13.97 |
| Pure Lowrank 8 with Iteration=10 | 9.84 | 20.10 | 0.763 | 0.362 | 34.53 | 13.87 |

Compact-Lowrank uses rank 32 and applies INT4 quantization to the transmitted $U$ and $V$, matching the communication cost of rank-8 FP16 methods. In contrast, the Pure Lowrank baselines use lower ranks (8 or 12) and do not quantize. As shown in Table 3, increasing the subspace iteration steps (from 2 to 10) brings negligible quality gains, while increasing the rank significantly improves generation quality. This highlights the importance of directional coverage over projection precision under tight bandwidth budgets.

## F.2   Effect of the Error Feedback

Table 4: Quantitative evaluation for Compact-1bit with and without Error Feedback mechanism.

| Method | PSNR (↑) | | LPIPS (↓) | | FID (↓) | |
|---|---|---|---|---|---|---|
| | w/ G.T. | w/ Orig. | w/ G.T. | w/ Orig. | w/ G.T. | w/ Orig. |
| Compact-1bit (with Error Feedback) | 9.80 | 22.90 | 0.767 | 0.260 | 33.20 | 7.08 |
| Compact-1bit (without Error Feedback) | 9.57 | 19.78 | 0.759 | 0.389 | 37.98 | 19.23 |

The results (Table 4) show that incorporating error feedback significantly improves performance across all metrics. Specifically, the Compact-1bit with Error Feedback variant achieves higher PSNR scores, lower LPIPS values, and notably better FID scores compared to the version without error feedback. This indicates that error feedback helps preserve perceptual and structural quality, making it a critical component for effective low-bit communication.

## F.3   Effect of the Warmup steps

Table 5: Quantitative evaluation for WarmUp Steps

| Method | PSNR (↑) | LPIPS (↓) | | FID (↓) | | Latency (s) | |
|---|---|---|---|---|---|---|---|
| | w/ Orig. | w/ G.T. | w/ Orig. | w/ G.T. | w/ Orig. | L20 | H20 |
| DistriFusion WarmUp 1 | 21.62 | 0.760 | 0.309 | 33.12 | 9.91 | 8.05 | 6.86 |
| DistriFusion WarmUp 2 | 23.37 | 0.762 | 0.261 | 33.11 | 8.46 | 8.38 | 6.93 |
| Compact-1bit WarmUp 1 | 22.89 | 0.766 | 0.259 | 33.20 | 7.08 | 7.46 | 6.86 |
| Compact-1bit WarmUp 2 | 24.48 | 0.768 | 0.219 | 33.26 | 6.23 | 7.63 | 6.92 |
| Compact-2bit WarmUp 1 | 29.54 | 0.772 | 0.114 | 33.09 | 3.26 | 7.57 | 6.70 |
| Compact-2bit WarmUp 2 | 31.02 | 0.772 | 0.095 | 33.11 | 2.78 | 7.71 | 6.89 |

Table 5 demonstrates that longer warm-up phases and higher-bit compression with error feedback improve visual quality.

Table 6: Human evaluation results. "Top-2 Count" refers to how often a method's output was selected as one of the *most similar* images; "Bottom-2 Count" refers to how often it was selected as one of the *least similar* ones. "Final Score" reflects the probability of being placed in the more consistent group.

| | PipeFusion | DistriFusion | Compact-Lowrank | Compact-1bit | Compact-2bit |
|---|---|---|---|---|---|
| Top-2 Count (↑) | 150 | 24 | 65 | 103 | **258** |
| Top-2 Rate (↑) | 0.50 | 0.08 | 0.22 | 0.34 | **0.86** |
| Bottom-2 Count (↓) | 113 | 173 | 139 | 119 | **56** |
| Bottom-2 Rate (↓) | 0.38 | 0.58 | 0.46 | 0.40 | **0.18** |
| Final Score (↑) | 0.56 | 0.25 | 0.38 | 0.47 | **0.84** |

### F.4  Details on Human Evaluation Results

Compact-2bit method can exhibit high consistency and fidelity with regard to human perception (Table 6).

### F.5  Temporal Stability Evaluation

Table 7: Temporal stability evaluation on CogVideoX (6 GPUs) following VBench [31] metrics.

| Method | Temporal Flickering (↑) | Motion Smoothness (↑) |
|---|---|---|
| Original (No Compression) | 0.9721 | 0.9817 |
| DistriFusion | 0.9658 | 0.9795 |
| Compact-2bit | 0.9644 | 0.9808 |
| Compact-1bit | 0.9661 | 0.9806 |
| Compact-Lowrank | 0.9623 | 0.9775 |

The results (Table 7) show that all CompactFusion variants maintain temporal stability comparable to or better than prior methods. The differences across metrics are minimal, indicating that compression introduces no noticeable degradation in temporal stability.

## G  More Qualitative Results

This section provides more qualitative results for FLUX.1-dev model and CogVideoX-2b model.

### G.1  FLUX.1-dev Qualitative Results

The supplementary qualitative results are shown in Table 8.

### G.2  CogVideoX-2b Qualitative Results

The supplementary qualitative results are shown in Table 9 and Table 10.

| Sequence Parallelism | DistriFusion | PipeFusion | Compact-1bit | Compact-2bit | Compact-Lowrank |
|---|---|---|---|---|---|

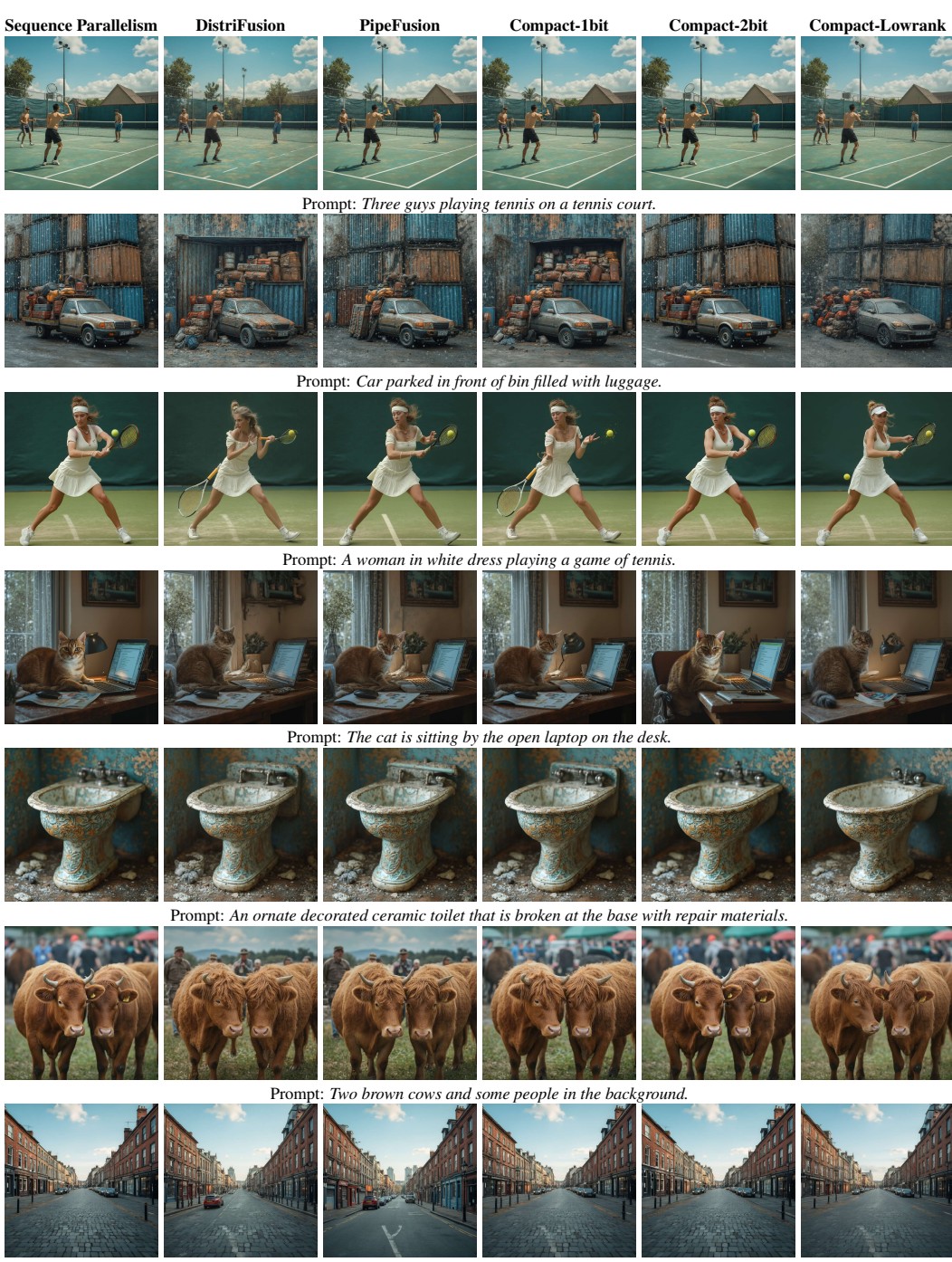

Prompt: *Three guys playing tennis on a tennis court.*

Prompt: *Car parked in front of bin filled with luggage.*

Prompt: *A woman in white dress playing a game of tennis.*

Prompt: *The cat is sitting by the open laptop on the desk.*

Prompt: *An ornate decorated ceramic toilet that is broken at the base with repair materials.*

Prompt: *Two brown cows and some people in the background.*

Prompt: *An empty street with brick buildings and cars.*

Table 8: Supplement Image Qualitative Results.

| Sequence Parallelism | DistriFusion | Compact-1bit | Compact-2bit | Compact-Lowrank |
|---|---|---|---|---|

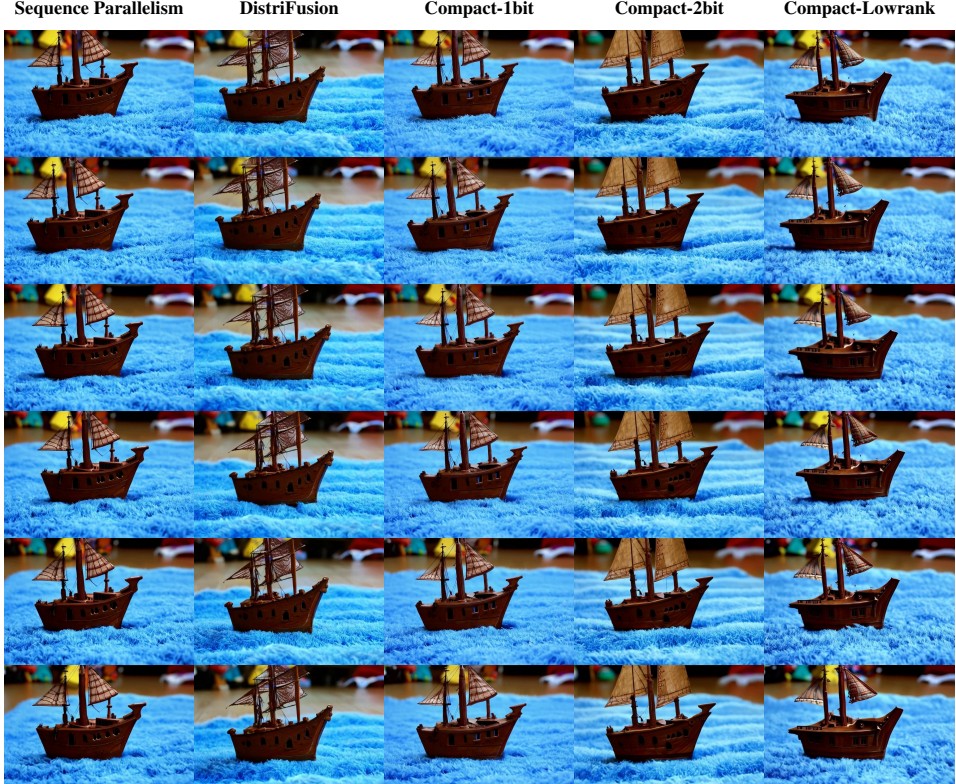

*Prompt: A detailed wooden toy ship with intricately carved masts and sails is seen gliding smoothly over a plush, blue carpet that mimics the waves of the sea. The ship's hull is painted a rich brown, with tiny windows. The carpet, soft and textured, provides a perfect backdrop, resembling an oceanic expanse. Surrounding the ship are various other toys and children's items, hinting at a playful environment. The scene captures the innocence and imagination of childhood, with the toy ship's journey symbolizing endless adventures in a whimsical, indoor setting.*

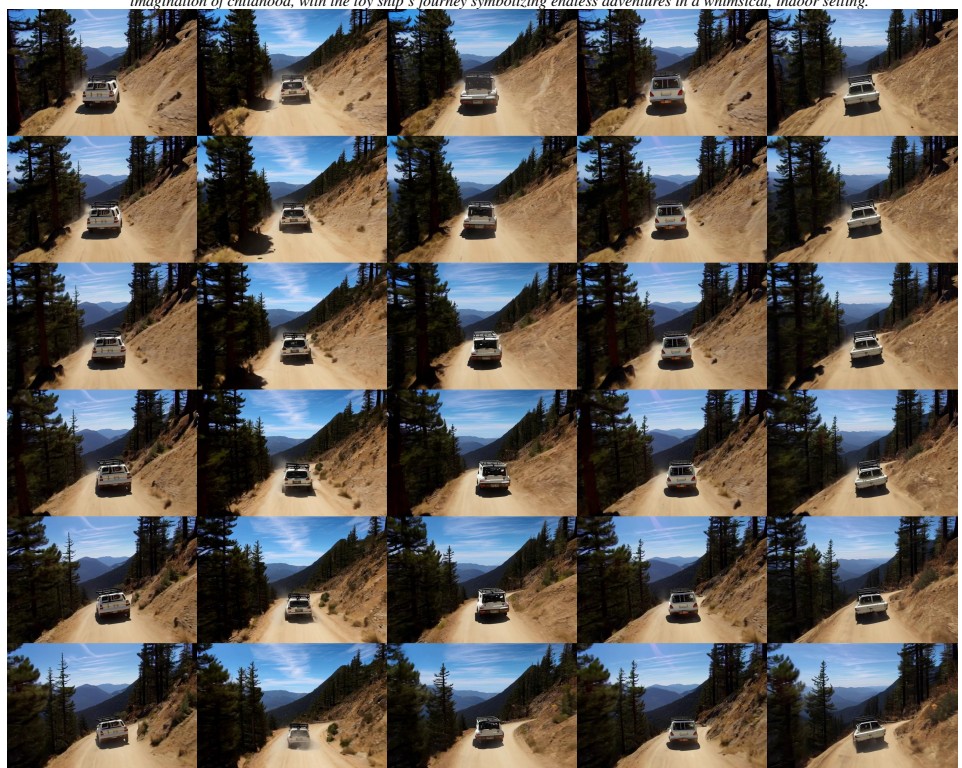

*Prompt: The camera follows behind a white vintage SUV with a black roof rack as it speeds up a steep dirt road surrounded by pine trees on a steep mountain slope, dust kicks up from it's tires, the sunlight shines on the SUV as it speeds along the dirt road, casting a warm glow over the scene. The dirt road curves gently into the distance, with no other cars or vehicles in sight. The trees on either side of the road are redwoods, with patches of greenery scattered throughout. The car is seen from the rear following the curve with ease, making it seem as if it is on a rugged drive through the rugged terrain. The dirt road itself is surrounded by steep hills and mountains, with a clear blue sky above with wispy clouds.*

Table 9: Supplement Video Qualitative Results (1/2).

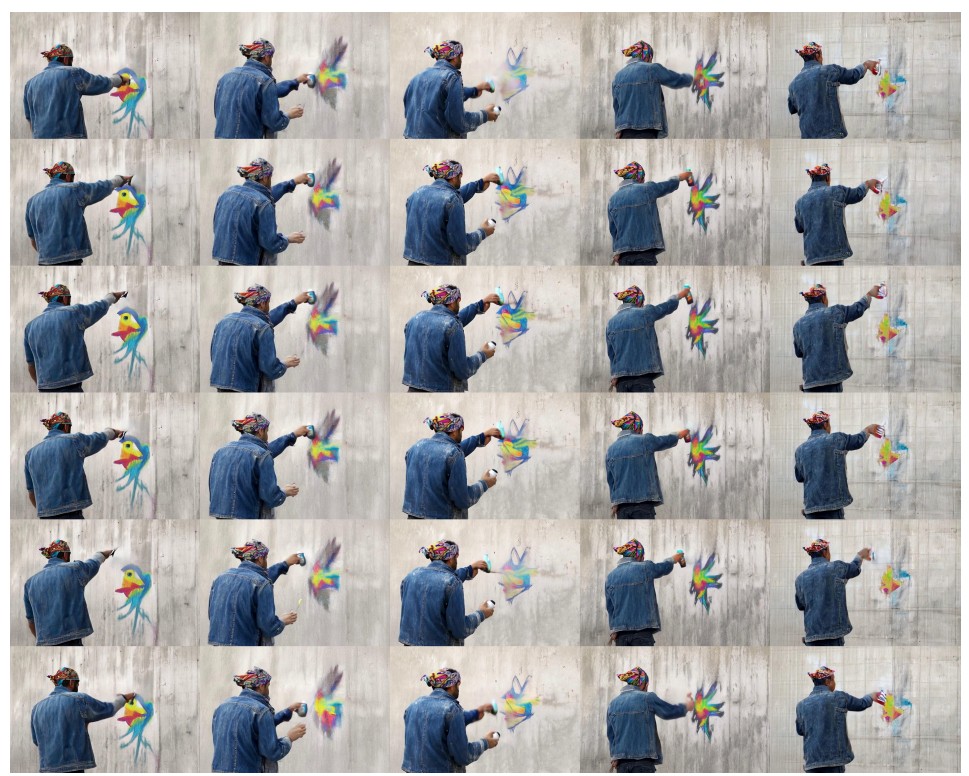

*Prompt: A street artist, clad in a worn-out denim jacket and a colorful bandana, stands before a vast concrete wall in the heart, holding a can of spray paint, spray-painting a colorful bird on a mottled wall.*

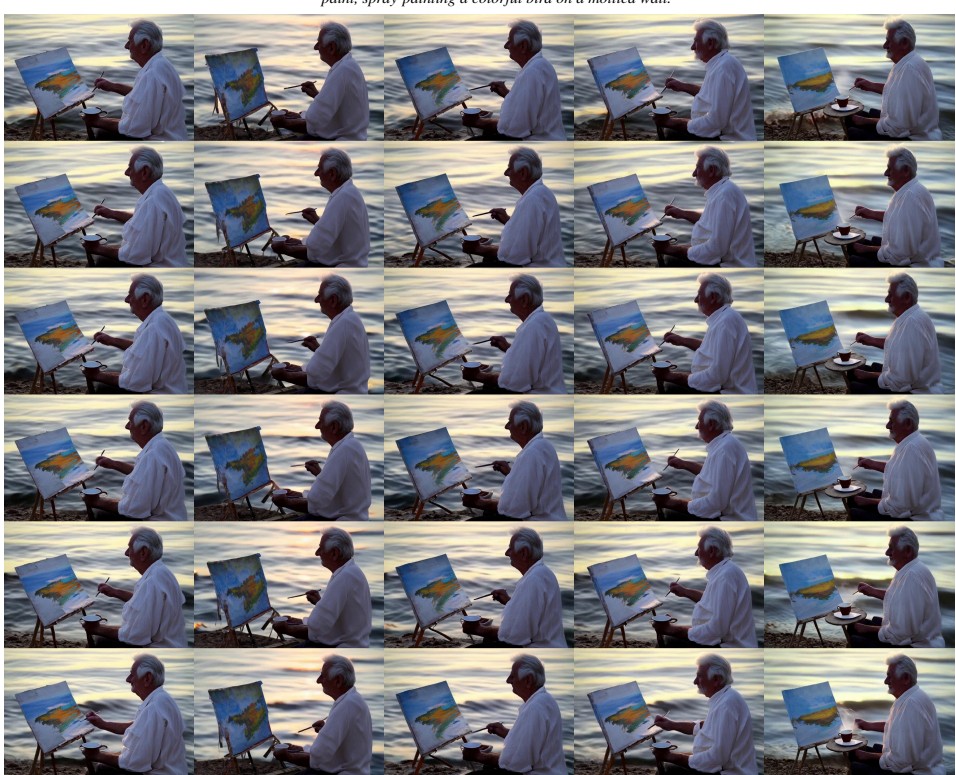

*Prompt: An elderly gentleman, with a serene expression, sits at the water's edge, a steaming cup of tea by his side. He is engrossed in his artwork, brush in hand, as he renders an oil painting on a canvas that's propped up against a small, weathered table. The sea breeze whispers through his silver hair, gently billowing his loose-fitting white shirt, while the salty air adds an intangible element to his masterpiece in progress. The scene is one of tranquility and inspiration, with the artist's canvas capturing the vibrant hues of the setting sun reflecting off the tranquil sea.*

Table 10: Supplement Video Qualitative Results (2/2).

## H  Compression Error Analysis

### H.1  Analysis Overview

This section provides detailed derivations for the steady-state expected squared error bounds for the Naive Compression and Residual Compression schemes discussed in the main paper. We also analyze a hypothetical residual compression scheme without error feedback to highlight the importance of the feedback mechanism.

### H.2  Definitions

We consider a function $f_t$ as a mapping from the previous state at step $t-1$ to the current state at step $t$. Let $a_t$ denote the true activation at step $t$, $\tilde{a}_t$ denote the reconstructed activation after compression at step $t$ and $a_t^*$ denote the activation get using the reconstructed activation from last step. All of these definitions target one specific layer. When warmup step is set to 1, the first step $\tilde{a}_1 = a_1$. $\{\tilde{a}_t\}_{t=1}^T$ will be the base that is stored locally in the case of residual compression. The generatl relationship can be expressed as

$$a_t = f_t(a_{t-1})$$

and

$$a_t^* = f_t(\tilde{a}_{t-1}), \ a_t^* \xrightarrow{\text{compress \& decompress}} \tilde{a}_t.$$

Then, we can use $\{a_t\}_{t=0}^T$ to denote an ideal, uncompressed sequence that is generated iteratively by the above function.

The total error is defined as the sum of the compression error and the propagation error

$$\tilde{e}_t = \tilde{a}_t - a_t = (\tilde{a}_t - a_t^*) + (a_t^* - a_t) = e_t + \eta_t,$$

where $e_t = \tilde{a}_t - a_t^*$ is the compression error and $\eta_t = a_t^* - a_t = f_t(\tilde{a}_{t-1}) - f_t(a_{t-1})$ is the propagation error. In the following derivation, $\Delta a_t = a_t - a_{t-1}$ is used to refer to the residual of true activation.

### H.3  Assumptions

We adopt the following assumptions for our error bound analysis:

**Assumption 1** ($L$-Smoothness). The function $f_t$ is $L$-smooth with $L < 1$

$$\|f_t(x) - f_t(y)\|^2 \leq L^2 \|x - y\|^2 \quad \forall x, y,$$

which implies that the process contracts over time, ensuring stability.

**Assumption 2** ($\delta$-Compressor). The compressor $C_\delta$ satisfies

$$\mathbb{E}\big[\|C_\delta(x) - x\|^2\big] \leq (1 - \delta)\mathbb{E}\big[\|x\|^2\big], \quad \delta \in (0, 1],$$

where $\delta$ measures the compression quality (higher $\delta$ indicates better fidelity).

**Assumption 3** (Bounded True Residual Variance). The true residuals have bounded expected energy:

$$\mathbb{E}\big[\|\Delta a_t\|^2\big] \leq \sigma_\Delta^2,$$

where $\sigma_\Delta^2$ is a finite constant.

**Assumption 4** (Bounded Activation Variance). The target activations have bounded expected values:

$$\mathbb{E}\big[\|a_t^*\|^2\big] \leq \sigma_a^2,$$

where $\sigma_a^2$ is a finite constant.

**Assumption 5** (Uncorrelated Errors (Simplification)). We assume certain terms are uncorrelated in expectation:

- $\mathbb{E}[\langle \Delta a_t, \tilde{e}_{t-1} \rangle] = 0$ (true residual uncorrelated with previous total error),
- $\mathbb{E}[\langle \delta f_t, \Delta a_t \rangle] = 0$ (perturbation effect uncorrelated with true residual),

- $\mathbb{E}[\langle \delta f_t, \tilde{e}_{t-1} \rangle] = 0$ (perturbation effect uncorrelated with previous total error),

- Compression error and propagation error are uncorrelated.

Here, $\delta f_t = f_t(a_{t-1} + \tilde{e}_{t-1}) - f_t(a_{t-1})$ is the perturbation effect due to the error in the input to $f_t$

## H.4    Analysis Model and Scope

Throughout this analysis, we model the evolution of the system state using the reconstructed state from the previous step, i.e., $a_t^* = f_t(\tilde{a}_{t-1})$. This means the function $f_t$ always operates on the potentially noisy state resulting from previous compression steps.

In practical parallel implementations, such as sequence parallelism, a device might compute attention using a combination of locally available, uncompressed ("fresh") Key/Value pairs and received Key/Value pairs that have undergone compression/decompression. Our analysis, by assuming the entire input context $\tilde{a}_{t-1}$ has passed through the compression cycle, simplifies the setup and likely provides a conservative upper bound on the error compared to such mixed-context scenarios. The core principles of error propagation and the benefits of compressing smaller residuals, however, remain applicable.

## H.5    Error Analysis for Naive Compression

**Proposition H.1.** *Let $v^{naive}$ denote the steady-state mean squared error upper bound under naive compression. Assuming the process has reached stationarity, the error satisfies the bound*

$$v^{naive} = \frac{(1-\delta)\sigma_a^2}{1 - L^2}, \quad provided \ L < 1.$$

*Proof.*

In the naive scheme, $\tilde{a}_t = C_\delta(a_t^*)$. The total error $\tilde{e}_t = \tilde{a}_t - a_t$ is decomposed into compression error $e_t^{naive} = \tilde{a}_t - a_t^*$ and propagation error $\eta_t = a_t^* - a_t = f_t(\tilde{a}_{t-1}) - f_t(a_{t-1})$. Assuming these errors are uncorrelated (Assumption 5):

$$\mathbb{E}[\|\tilde{e}_t\|^2] = \mathbb{E}[\|e_t^{naive}\|^2] + \mathbb{E}[\|\eta_t\|^2].$$

Using the $\delta$-compressor property (Assumption 2) and the bounded activation variance assumption (Assumption 4), we can bound the compression error by

$$\mathbb{E}[\|e_t^{naive}\|^2] = \mathbb{E}[\|C_\delta(a_t^*) - a_t^*\|^2] \le (1-\delta)\mathbb{E}[\|a_t^*\|^2] \le (1-\delta)\sigma_a^2.$$

Using the $L$-smoothness assumption (Assumption 1), we can bound the propagation error by

$$\mathbb{E}[\|\eta_t\|^2] = \mathbb{E}[\|f_t(\tilde{a}_{t-1}) - f_t(a_{t-1})\|^2] \le L^2 \mathbb{E}[\|\tilde{a}_{t-1} - a_{t-1}\|^2] = L^2 \mathbb{E}[\|\tilde{e}_{t-1}\|^2].$$

Combining the terms yields the recurrence

$$\mathbb{E}[\|\tilde{e}_t\|^2] \le (1-\delta)\sigma_a^2 + L^2 \mathbb{E}[\|\tilde{e}_{t-1}\|^2].$$

Letting $v^{naive}$ as the steady-state error upper bound and assuming $\mathbb{E}[\|\tilde{e}_t\|^2] = \mathbb{E}[\|\tilde{e}_{t-1}\|^2]$ in steady state, we obtain:

$$v^{naive} = (1-\delta)\sigma_a^2 + L^2 v^{naive}.$$

Solving for $v^{naive}$ gives the bound

$$v^{naive} = \frac{(1-\delta)\sigma_a^2}{1 - L^2} \quad (\text{requires } L < 1),$$

where stability requires $L < 1$. $\qquad\qquad\square$

## H.6 Error Analysis for Residual Compression

**Proposition H.2.** *Let $v^{residual}$ denote the steady-state mean squared error upper bound for residual compression. Assuming the process has reached stationarity, the error is bounded by*

$$v^{residual} = \frac{(1-\delta)\sigma_\Delta^2}{1 - L^2 - (1-\delta)(L^2+1)}, \quad provided\ L^2 + (1-\delta)(L^2+1) < 1.$$

*Proof.*

In the residual scheme, we let $\Delta a_t^* = a_t^* - \tilde{a}_{t-1}$, $\tilde{\Delta} a_t = C_\delta(\Delta a_t^*)$, and $\tilde{a}_t = \tilde{a}_{t-1} + \tilde{\Delta} a_t$. The total error is $\tilde{e}_t = e_t^{\text{residual}} + \eta_t$, where $e_t^{\text{residual}} = \tilde{a}_t - a_t^* = \tilde{\Delta} a_t - \Delta a_t^*$. Follow the assumption that $e_t^{\text{residual}}$ and $\eta_t$ are uncorrelated (Assumption 5), we can get

$$\mathbb{E}[\|\tilde{e}_t\|^2] = \mathbb{E}[\|e_t^{\text{residual}}\|^2] + \mathbb{E}[\|\eta_t\|^2].$$

Using the $\delta$-compressor property (Assumption 2):

$$\mathbb{E}[\|e_t^{\text{residual}}\|^2] \le (1-\delta)\mathbb{E}[\|\Delta a_t^*\|^2].$$

Since

$$\Delta a_t^* = f_t(\tilde{a}_{t-1}) - \tilde{a}_{t-1} = [f_t(a_{t-1}+\tilde{e}_{t-1}) - f_t(a_{t-1})] + [f_t(a_{t-1}) - a_{t-1}] - \tilde{e}_{t-1} = \delta f_t + \Delta a_t - \tilde{e}_{t-1},$$

where $\delta f_t = f_t(a_{t-1} + \tilde{e}_{t-1}) - f_t(a_{t-1})$ and $\Delta a_t = a_t - a_{t-1} = f_t(a_{t-1}) - a_{t-1}$. Applying the uncorrelatedness assumptions (Assumption 5), we can further bound $\mathbb{E}[\|\Delta a_t^*\|^2]$

$$\begin{aligned}
\mathbb{E}[\|\Delta a_t^*\|^2] &= \mathbb{E}[\|\delta f_t\|^2] + \mathbb{E}[\|\Delta a_t\|^2] + \mathbb{E}[\|\tilde{e}_{t-1}\|^2] \\
&\le L^2 \mathbb{E}[\|\tilde{e}_{t-1}\|^2] + \sigma_\Delta^2 + \mathbb{E}[\|\tilde{e}_{t-1}\|^2] \quad \text{(using Assumption 1 and Assumption 3)} \\
&= (L^2+1)\mathbb{E}[\|\tilde{e}_{t-1}\|^2] + \sigma_\Delta^2.
\end{aligned}$$

Thus, the compression error is bounded by

$$\mathbb{E}[\|e_t^{\text{residual}}\|^2] \le (1-\delta)\big[(L^2+1)\mathbb{E}[\|\tilde{e}_{t-1}\|^2] + \sigma_\Delta^2\big].$$

The propagation error term is bounded exactly as in the naive case: $\mathbb{E}[\|\eta_t\|^2] \le L^2\mathbb{E}[\|\tilde{e}_{t-1}\|^2]$.

Combining the bounds for compression and propagation errors:

$$\mathbb{E}[\|\tilde{e}_t\|^2] \le (1-\delta)\big[(L^2+1)\mathbb{E}[\|\tilde{e}_{t-1}\|^2] + \sigma_\Delta^2\big] + L^2\mathbb{E}[\|\tilde{e}_{t-1}\|^2].$$

Letting $v^{\text{residual}}$ as the steady-state error upper bound and assuming $\mathbb{E}[\|\tilde{e}_t\|^2] = \mathbb{E}[\|\tilde{e}_{t-1}\|^2]$ in steady state, we obtain:

$$v^{\text{residual}} = (1-\delta)(L^2+1)v^{\text{residual}} + (1-\delta)\sigma_\Delta^2 + L^2 v^{\text{residual}}.$$

Solving yields the closed-form upper bound:

$$v^{\text{residual}} = \frac{(1-\delta)\sigma_\Delta^2}{1 - L^2 - (1-\delta)(L^2+1)},$$

where stability requires:

$$1 - L^2 > (1-\delta)(L^2+1).$$

## H.7 Analysis Without Error Feedback

Consider a hypothetical scheme where the true residual $\Delta a_t = a_t - a_{t-1}$ is compressed and added to the previous reconstruction: $\tilde{a}_t = \tilde{a}_{t-1} + C_\delta(\Delta a_t)$. The error $\tilde{e}_t = \tilde{a}_t - a_t$ evolves as:

$$\tilde{e}_t = (\tilde{a}_{t-1} + C_\delta(\Delta a_t)) - (a_{t-1} + \Delta a_t) = (\tilde{a}_{t-1} - a_{t-1}) + (C_\delta(\Delta a_t) - \Delta a_t) = \tilde{e}_{t-1} + \epsilon_t,$$

where $\epsilon_t = C_\delta(\Delta a_t) - \Delta a_t$ is the compression error on the true residual. If we assume $\epsilon_t$ are uncorrelated over time, the expected squared error (variance) accumulates:

$$\mathbb{E}[\|\tilde{e}_t\|^2] = \mathbb{E}[\|\tilde{e}_{t-1}\|^2] + \mathbb{E}[\|\epsilon_t\|^2] \le \mathbb{E}[\|\tilde{e}_{t-1}\|^2] + (1-\delta)\sigma_\Delta^2.$$

This indicates that $\mathbb{E}[\|\tilde{e}_t\|^2]$ grows linearly with $t$ and does not converge to a finite steady state, highlighting the necessity of the error feedback mechanism (implicit in using $\tilde{a}_{t-1}$ to compute the target $a_t^*$ and the residual $\Delta a_t^*$) for stability.

### H.8 Steady-State Error Bounds

We present the derived steady-state expected squared error bounds for naive compression ($v^{\text{naive}}$) and residual compression with error feedback ($v^{\text{residual}}$), based on the given assumptions. The results are as follows.

$$v^{\text{naive}} = \frac{(1-\delta)\sigma_a^2}{1-L^2}, \quad v^{\text{residual}} = \frac{(1-\delta)\sigma_\Delta^2}{1-L^2-(1-\delta)(L^2+1)}.$$

Provided the stability condition for residual compression is satisfied and given that, typically, $\sigma_\Delta^2 \ll \sigma_a^2$, the residual scheme offers a lower theoretical error bound (although a precise comparison requires evaluating all expressions, including denominators).

## I License and Asset Attribution

We use two open-source implementations in our experiments:

- **xDiT(xfuser)** [10]: A parallel diffusion inference framework released under the Apache-2.0 license.
- **DistriFusion(distrifuser)** [18]: Released under the MIT license.

Both projects are publicly available, and we have properly credited their creators in the main text. We have respected all terms of their licenses, and all usage is compliant with their respective open-source agreements. No proprietary assets or restricted-use models were used in this work.

All remaining components of CompactFusion, including our compression modules and integration layers, are implemented independently.

## J Broader Impacts

CompactFusion improves the efficiency of diffusion model inference by reducing communication overhead. This can lower deployment cost and enable generative models to run on a wider range of hardware.

These improvements may help democratize access to generative AI and reduce energy consumption in large-scale deployments. We do not foresee negative societal impacts from this work.

