# OpenReview forum: "Accelerating Parallel Diffusion Model Serving with Residual Compression"
_NeurIPS.cc/2025/Conference — NeurIPS 2025 poster_

### Official Review · Reviewer_QZGw · 2025-06-27

**Clarity:** 3
**Significance:** 3
**Originality:** 3
**Rating:** 5
**Confidence:** 3

**Summary:**

Diffusion models are computationally intensive and require multi-accelerator parallelism for real-time deployment, but suffer from high communication overhead due to redundant activation exchanges. This paper propose to addresses this by introducing a Residual Compression technique that transmits only compressed residuals, effectively reducing redundancy. This method maintains high generation quality while significantly lowering latency than prior methods.

**Questions:**

- It would be beneficial to include an additional ablation study comparing the model's performance with and without quantization errors. This analysis would help isolate the impact of compression-induced errors and provide a clearer understanding of how quantization affects the overall performance. It may also help understand the upper bound of the effectiveness of the proposed error feedback.
- Has the influence of content complexity been examined? It would be helpful to analyze whether the proposed method performs differently on inputs with complex patterns versus simpler ones.
- The proposed method introduces compression errors. Will it affect the temporal stability of the generated videos. Has this aspect been evaluated?

**Ethical Concerns:**

["NO or VERY MINOR ethics concerns only"]

**Final Justification:**

This paper effectively reduces redundancy while maintaining high generation quality. The authors have provided a clear and satisfactory rebuttal that addresses my previous concerns. Given that, I raise my score.

**Limitations:**

yes

**Paper Formatting Concerns:**

I did not find major formatting issues.

**Quality:**

3

**Strengths And Weaknesses:**

- Strengths
  - The proposed CompactFusion delivers lower latency parallel diffusion inference, while achieves higher generation quality than prior methods.
  - Both empirical analysis and theoretical justification are given to show the effectiveness of the proposed method.
- Weaknesses
  - I do not see obvious weaknesses. Some small issues are mentioned in Questions.

---

> ### Author Rebuttal · Authors · 2025-07-28
>
> ## Thank you for your helpful feedback!
>
> We sincerely thank reviewer QZGw for the positive assessment of our work. We are glad that the strengths of **low-latency, high-quality parallel diffusion inference**, as well as the combination of **empirical and theoretical validation**, were recognized. The reviewer also raised several constructive questions, and we address them individually below.
>
> ---
> # Ablation: Impact of Quantization
>
> > It would be beneficial to include an additional ablation study comparing the model's performance with and without quantization errors. This analysis would help isolate the impact of compression-induced errors and provide a clearer understanding of how quantization affects the overall performance. It may also help understand the upper bound of the effectiveness of the proposed error feedback.
>
> **We thank the reviewer for this helpful suggestion. We agree that isolating the impact of quantization and the benefit of error feedback is valuable. In fact, we have already conducted such an ablation, comparing the following:**
>
> - No Compression (baseline)
> - 1-bit quantization only
> - 1-bit quantization with error feedback
>
> The baseline numbers (no compression) are included in Table 1 of the main paper, and the rest are reported in Appendix E.2. For convenience, we summarize results below:
>
> **Table: Ablation Study on Quantization and Error Feedback (28 steps, FLUX, 4×L20)**
>
> Note: w/ G.T. refers to metrics computed against ground truth COCO images, while w/ Orig. compares to outputs from the original (uncompressed) model.
>
> | Method                            | Description                            | LPIPS ↓ (w/ G.T.) | LPIPS ↓ (w/ Orig.) | FID ↓ (w/ G.T.) | FID ↓ (w/ Orig.) |
> |----------------------------------|----------------------------------------|-------------------|--------------------|------------------|------------------|
> | Original (No Compression)  | Without quantization error       | 0.772   | –  | 32.75    | – |
> | Compact-1bit (w/o Error Feedback)| With quantization only                 | 0.759             | 0.389              | 37.98            | 19.23            |
> | Compact-1bit (w/ Error Feedback) | With quantization and error feedback   | 0.767             | 0.260              | 33.20            | 7.08             |
>
> **Quantization alone introduces significant degradation (FID 32.75 -> 37.98), but error feedback effectively mitigates it (FID 37.98 -> 33.20).**
>
> # Content Complexity Does Not Affect Our Method
>
> > Has the influence of content complexity been examined? It would be helpful to analyze whether the proposed method performs differently on inputs with complex patterns versus simpler ones.
>
> Thank you for raising this point.
>
> In current state-of-the-art diffusion models like FLUX, the input is textual and padded to a fixed length. As a result, the complexity of the input text has negligible effect on runtime or parallel behavior. What actually affects performance is the resolution of the generated image or video—not the complexity of the prompt.
>
> **Therefore, content complexity does not impact our method, and to our knowledge, prior works such as PipeFusion and DistriFusion have also not addressed this aspect. That said, we appreciate the reviewer for bringing up this dimension, and we will clarify this point in Appendix B: Discussion in the revision.**
>
> # Compression Does Not Affect Temporal Stability
> > The proposed method introduces compression errors. Will it affect the temporal stability of the generated videos. Has this aspect been evaluated?
>
> We thank the reviewer for raising this point regarding whether compression might harm the temporal consistency of generated videos.
>
> Like prior works such as DistriFusion and PipeFusion, we did not originally include explicit temporal stability metrics. However, we agree this is a helpful addition.
>
> We followed the VBench[1] to evaluate two standard metrics:
>
> - **Temporal Flickering** (higher is better): Measures brightness/color jitter between consecutive frames.
> - **Motion Smoothness** (higher is better): Measures motion consistency based on optical flow.
>
> The setup follows Section 4.2 (CogVideoX, 6 GPUs).
>
> **Table: Temporal Stability Evaluation (VBench, higher = better)**
>
> | Method                 | Temporal Flickering ↑ | Motion Smoothness ↑ |
> |------------------------|------------------------|----------------------|
> | Original (No Compression)  | 0.9721                 | 0.9817               |
> | DistriFusion           | 0.9658                 | 0.9795               |
> | Compact-2bit           | 0.9644                 | 0.9808               |
> | Compact-1bit           | 0.9661                 | 0.9806               |
> | Compact-Lowrank       | 0.9623                 | 0.9775               |
>
> The differences across methods are minimal, showing that **compression causes no or negligible degradation in temporal stability**.
>
> **We thank the reviewer again for highlighting this aspect, and we will include these results in Appendix E: More Quantitative  Results in the revision.**
>
> ---
>
> **References**
>
> *[1] Z. Huang et al., “VBench: Comprehensive Benchmark Suite for Video Generative Models,” in 2024 IEEE/CVF Conference on Computer Vision and Pattern Recognition (CVPR), Seattle, WA, USA: IEEE, June 2024, pp. 21807–21818. doi: 10.1109/CVPR52733.2024.02060.*

---

> > ### Comment · Reviewer_QZGw · 2025-08-04
> >
> > Thanks for the high-quality feedback given by the authors, which have addressed my concerns.

---

> > > ### Author Response · Authors · 2025-08-06
> > >
> > > Thanks for your recognition.
> > >
> > > Hope all’s going well:)

---

### Official Review · Reviewer_ht4v · 2025-06-29

**Clarity:** 2
**Significance:** 2
**Originality:** 3
**Rating:** 4
**Confidence:** 2

**Summary:**

To reduce communication overhead in patch parallel inference, this work proposes a novel Fresh Activation strategy. The method yields a modest speedup while preserving model performance through an error feedback mechanism.

**Questions:**

please see the weekness

**Ethical Concerns:**

["NO or VERY MINOR ethics concerns only"]

**Final Justification:**

Thanks to the author for the Rebuttal.
I have read it carefully. My concerns are well addressed and I will raise my score.

**Limitations:**

yes

**Quality:**

3

**Strengths And Weaknesses:**

[Strengthen]
1.The proposed method improves the efficiency of parallel inference by reusing activations through the Fresh mechanism.
2.The authors provide a theoretical analysis demonstrating that the Fresh activation reuse effectively reduces the approximation error.
3.The method is integrated with existing toolchains, showing potential for practical deployment in real-world applications.

[Weakness]
1. The motivation of the work lacks clarity. Since communication and computation are overlapped in practice, compressing activations is typically aimed at reducing communication latency. This motivation implicitly assumes that communication latency completely dominates computation time. The paperdoes not provide any empirical evaluation or profiling to justify this assumption.
2. The effectiveness of activation compression in improving inference speed remains questionable. DistriFusion adopts a longer warm-up process, while CompactFusion only uses a one-step warm-up, which inherently gives it an advantage in speed. However, as shown in Tables 1 and 2, CompactFusion only yields marginal improvements over DistriFusion, especially when low-rank compression is applied. This raises concerns that transmitting compressed activations may not meaningfully accelerate inference.
3. The implementation details of patch parallelism in CompactFusion are unclear. Does it appear to be built upon the parallelism strategy of DistriFusion, with additional transmission of the residual for the current patch? The paperlacks a clear description of this strategy, leading to confusion. For example, in Figure 3, what does “base” refer to? Why does the error feedback correct the sender’s “base” instead of that of the receiver? How are all layers of the model executed during parallel inference? A more explicit explanation of the overall parallel procedure is necessary.
4. The paperclaims that CompactFusion generalizes better than DistriFusion across model architectures and inference strategies. However, the underlying reasons for this improved generalization are not clearly articulated. A detailed analysis of the compatibility mechanisms employed by CompactFusion is needed to substantiate this claim.

---

> ### Author Rebuttal · Authors · 2025-07-27
>
> ## We sincerely appreciate your feedback and questions!
>
> We thank reviewer ht4v for recognizing the strengths of our work, including its **efficiency gains, theoretical support, and potential for practical deployment**.
>
> We also thank the reviewer for raising several insightful points—especially on motivation, performance, and generalization.
>
> We believe our responses below provide strong evidence that CompactFusion addresses a real and pressing challenge with solid performance advantages.
> For the concerns on preliminaries, we promise to improve and expand them in the revision.
>
> ---
>
> # Communication Optimization Is Necessary in Practice
> > The motivation of the work lacks clarity. Since communication and computation are overlapped in practice, compressing activations is typically aimed at reducing communication latency. This motivation implicitly assumes that communication latency completely dominates computation time. The paper does not provide any empirical evaluation or profiling to justify this assumption.
>
> This is an excellent point—thank you for raising it. We’re glad to expand on it here.
>
> First, we would like to point out that current diffusion model parallelism typically exhibits no or only limited communication-computation overlap:
>
> - DeepSpeed-Ulysses uses synchronous all-to-all communication, **with no overlap**;
> - Patch Parallel uses synchronous all-gather, also **without overlap**;
> - Ring Attention does enable **limited overlap**—each ring-style transmission consists of multiple P2P steps, where each step can overlap partially with a small slice of the attention computation.
>
> Under these no/limited overlap designs, communication latency becomes a major bottleneck. For instance, even in Ring Attention over 4xL20, we observed that communication blocking accounts for **36%** of total latency in CogVideoX-2b, and **43%** in FLUX, despite that Ring Attention allows limited overlap. In fact, on FLUX, Ring Attention offers almost no improvement over Patch Parallel in practice.
>
> **We note that, contrary to the concern, Section 1 already includes profiling and real‑measurement data that support our motivation.**
> > Line 34: In FLUX.1, standard Patch Parallelism... consuming over **45%** of inference time … underscoring the need to reduce transmission overhead.
>
> **We have also noted that hardware trends would make the communication bottleneck even more critical over time in Section 1:**
> > Line32: interconnect bandwidth has not kept pace with the growth of the compute. From A100 to H100, FP16 FLOPS increases over **6×** (312T → 1,979T), while NVLink bandwidth grows only **1.5×** and PCIe bandwidth simply **doubles**.
>
> We appreciate the reviewer’s attention to this point, and we emphasize that the motivation is well grounded: **communication latency is substantial, especially in realistic deployment settings**. In fact, this is exactly why a line of prior work—including DistriFusion and PipeFusion—focuses on communication optimization. CompactFusion addresses the same problem but with a fundamentally different approach.
>
> # CompactFusion Achieves Much Better Quality—While Also Being Faster
> > The effectiveness of activation compression in improving inference speed remains questionable. DistriFusion adopts a longer warm-up process, while CompactFusion only uses a one-step warm-up, which inherently gives it an advantage in speed...
>
> We thank the reviewer for the opportunity to clarify the configuration and performance.
>
> In our experiments, we use **1-step warmup for all methods, including CompactFusion, DistriFusion, and PipeFusion. This ensures that comparisons are completely fair.** While Figure 1 and Figure 6 indicate the 1-step setting, we acknowledge that Tables 1 and 2 do not repeat this detail—we promise to clarify this in the revision.
>
> **We would also like to emphasize that CompactFusion achieves significantly better generation quality than DistriFusion—while also being faster.**
>
> Under PCIe on FLUX (Table 1), Compact-2bit achieves a FID of 3.26 vs. DistriFusion’s 9.91—**an over 3× improvement in quality metric, while also being faster** (7.57s vs. 8.05s). Similar trends are observed in CogVideoX.
>
> In slower networks, our advantage becomes even more pronounced. Under Ethernet-level bandwidth, CompactFusion achieves up to **6.7× speedup over DistriFusion** (Figure 7).
>
> CompactFusion delivers significant improvement over existing methods.
>
> # Clarification on Patch Parallelism Implementation
> > The implementation details of patch parallelism in CompactFusion are unclear. Does it appear to be built upon the parallelism strategy of DistriFusion, with additional transmission of the residual for the current patch?
>
> We thank the reviewer for highlighting this aspect, and we’re happy to elaborate.
> In our paper, Patch Parallel refers to a standard sequence parallelism strategy, where devices exchange Key/Value tensors via all-gather. This is the exact communication pattern used in DistriFusion (which builds displaced patch parallelism on top of it).
>
> In CompactFusion, we apply compression to this all-gather stage—so instead of transmitting full Key/Value tensors, we transmit compressed residuals. The underlying parallel strategy remains unchanged.
>
> We understand the reviewer’s concern. Since CompactFusion is **parallel-strategy agnostic**, we initially felt it was unnecessary to detail a specific parallel implementation. That said, we agree that providing this information will improve clarity.
>
> **To help clarify this, we promise to include an extra Preliminaries Section in the Appendix that will cover more background and procedural details.** Specifically, we will:
>
> - Include illustrative figures and pseudocode to show how parallel strategies (e.g., Patch Parallel) work in practice;
>
> - Provide a detailed explanation of the DistriFusion to help readers who may be unfamiliar with overlap-based methods better understand its mechanism.
>
> # Clarification on "base" and why error feedback occurs on the sender side
> > The paper lacks a clear description of this strategy, leading to confusion. For example, in Figure 3, what does “base” refer to? Why does the error feedback correct the sender’s “base” instead of that of the receiver? How are all layers of the model executed during parallel inference? A more explicit explanation of the overall parallel procedure is necessary.
>
> The reviewer raised a great question about the residual base and error feedback mechanism.
>
> In CompactFusion, we transmit residuals—i.e., the change in activation value since the last step. To compute a residual, **one must subtract a reference value—this is what we call the base**. The base is initialized with the uncompressed activation in the warm-up step, and updated over time with compensated error.
>
> When you compress a residual, you introduce an error. This compression error is only visible to the sender, because only the sender knows the true residual before compression. **The receiver, on the other hand, only sees the compressed result—it has no way of knowing how much error was introduced.**
>
> Therefore, error feedback can only be performed on the sender side. At each step, the sender adds back the previous error into the new residual, so that missing information is gradually recovered.
>
> We will add the clarification of this mechanism in Appendix C: Implementation Details in the revision.
>
> # CompactFusion Generalizes More Easily Across Both Models and Strategies
> > Why and how does CompactFusion generalize better than overlap-based methods(DistriFusion, PipeFusion) across models and parallel strategies?
>
> We’re glad the reviewer brought this up. Generalization is one of CompactFusion’s key strengths, and it is supported by our practical experience.
>
> ### Generalization Across Parallel Strategies
>
> As noted earlier, CompactFusion is agnostic to the specific parallel strategy. It does not modify the model or pipeline logic; instead, **it only wraps the communication calls**. This design allows it to work with more strategies.
>
> By contrast, overlap-based methods are tightly bound to specific parallel strategies:
>
> - DistriFusion is specific to Patch Parallel.
> - PipeFusion is specific to TeraPipe.
>
> ### Generalization Across Model Architectures
>
> **This parallel strategy compatibility directly translates into model-level portability.**
>
> PipeFusion and DistriFusion are tightly bound to specific parallel methods (e.g., TeraPipe, Patch Parallel), which are often not supported by a given model. As a result, applying them to a new model often requires building the entire parallel strategy from scratch.
>
> In contrast, CompactFusion is modular and does not rely on any specific method. We do not need to implement any specific parallel strategy from scratch. **Instead, we directly plug into the parallel strategy that the model already uses—in practice, this is almost always Ring Attention.**
>
> ### CompactFusion Requires Minimal Integration Effort
>
> To make the difference concrete, we illustrate using our two evaluated models, FLUX and CogVideoX. Here’s what it took to run each method in xDiT[1] (to the best of our knowledge, the most widely used diffusion parallel framework):
>
> - **PipeFusion** – CogVideoX does not have TeraPipe in xDiT. Implementing it requires extensive and complex re‑engineering (FLUX already has TeraPipe-style parallel from the xDiT team.)
>
> - **DistriFusion** – Neither model has Patch Parallel. We implemented it ourselves for both, mainly for experiments. Easier than TeraPipe, but still needs substantial efforts.
>
> - **CompactFusion – wrap the communication with compress&decompress in an already supported strategy (Ring Attention). Model-agnostic, <10 core lines of code.**
>
> ---
> **References**
>
> *[1] J. Fang, J. Pan, X. Sun, A. Li, and J. Wang, “xDiT: an Inference Engine for Diffusion Transformers (DiTs) with Massive Parallelism,” Nov. 04, 2024, arXiv: arXiv:2411.01738. doi: 10.48550/arXiv.2411.01738.*

---

> > ### Comment · Reviewer_ht4v · 2025-08-06
> >
> > Thanks to the author for the Rebuttal. I have read it carefully. My concerns are well addressed and I will raise my score.

---

> > > ### Author Response · Authors · 2025-08-06
> > >
> > > Glad we could address your concerns.
> > >
> > > Thanks again, and all the best ahead:)

---

### Official Review · Reviewer_z8iV · 2025-07-02

**Clarity:** 3
**Significance:** 2
**Originality:** 2
**Rating:** 4
**Confidence:** 4

**Summary:**

This work aims to accelerate parallel diffusion model inference by mitigating communication bottlenecks. The authors focus on the problem of excessive communication overhead caused by exchanging large activation data between devices during distributed training. They address limitations in prior approaches, such as quality degradation from stale activation reuse and insufficient reduction of data volume. The authors propose a compression method that transmits compressed residuals (differences between consecutive steps) instead of full activations, combined with lightweight error feedback to maintain fidelity. The proposed method integrates optimized GPU kernels and system co-design to overlap compression with communication. Evaluations on show that the proposed method brings inference speedups on PCIe/Ethernet with small quality loss and perceptual degradation.

**Questions:**

Please clarify my questions in the weakness part.

**Ethical Concerns:**

["NO or VERY MINOR ethics concerns only"]

**Final Justification:**

The authors have addressed my major concerns, so I will keep my rating.

**Limitations:**

This paper has some merits and I have summarized my major concerns in the weakness part.

**Paper Formatting Concerns:**

N/A.

**Quality:**

3

**Strengths And Weaknesses:**

**Strengths:**

The authors tackle the communication bottleneck in parallel diffusion model inference. Previous methods, such as reusing stale activations, reduce data volume but degrade generation quality. This paper has pointed out a fundamental challenge of balancing communication efficiency and perceptual quality in diffusion modeling.

The proposed method leverages residual compression, encoding only the differences between consecutive steps, instead of transmitting full activations between steps. This idea is simple but effective and has been validated by the experiments.


**Weaknesses:**

Mostly, I think this paper is interesting. Here are some aspects that could be strengthened.

The authors transmit compressed residuals rather than full activations and leverage temporal redundancy in diffusion steps. So, the compression operations may introduce additional latency. Although it can be mitigated by overlapping with communication and computations, I’m interested in how to balance this compression-latency problem.

---

> ### Author Rebuttal · Authors · 2025-07-26
>
> ## Thank you for your constructive comments!
> We sincerely thank reviewer z8iV for the thoughtful and positive feedback. We’re glad that the core contributions of our work were recognized—**identifying the quality-efficiency tradeoff in parallel diffusion**, **proposing an effective residual compression scheme**, and **validated by empirical results**.
>
> The reviewer also raised an insightful point regarding the potential latency introduced by compression. We address this concern in detail below.
>
> ---
>
> # The Compression Latency Is Indeed Challenging
>
> > The authors transmit compressed residuals rather than full activations and leverage temporal redundancy in diffusion steps. So, the compression operations may introduce additional latency. Although it can be mitigated by overlapping with communication and computations, I’m interested in how to balance this compression-latency problem.
>
> **The reviewer raised a great question about the latency overhead from compression.**
>
> While our paper briefly mentions some of the optimizations, we’re glad to have the opportunity to walk through the full picture—**this was actually one of the trickiest parts of the system.**
>
> Compression latency is a real issue. While communication contributes significantly to total inference time, it is highly fragmented, with each individual transfer being very short. On 4×L20 with Ring Attention, a single ring-style communication blocks for less than ~5ms. When compression is enabled, each communication involves one compression and three decompressions. To achieve any meaningful speedup, the **total time for all these operations(1x compress & 3x decompress) must be well under 5ms—leaving very little headroom for overhead.**
>
> # We Tackle the Latency Challenge through Multi-Level Co-Design and Engineering
>
> To meet that goal, we had to be careful at every level.
>
>  **First, we selected only compression strategies that are viable under such tight constraints.** CompactFusion supports sparsity, quantization, and low-rank—but not all variants are usable. We ruled out Top-k sparsity due to its O(n log k) complexity. For low-rank, we use subspace iteration instead of SVD, as SVD is far too slow (appendix C). For quantization, we avoid complex adaptive schemes and use fast, fixed-format quantizers.
>
> **Next came hardware considerations. Some methods that look promising on paper don’t work well on GPU.** Hard-thresholding(keeping all elements above a certain threshold), for instance, creates irregular sparsity patterns that are hard to encode, decode, and parallelize efficiently. Instead, we adopted N:M sparsity, which, while previously unused in activation compression, maps well to GPU hardware due to its block structure. **When we say in Section 3.4 that “it is the only sparsifier design we found to yield practical on-device speedup,” that’s backed by a lot of testing and design.**
>
>  **Even with the right algorithms, naive implementations were not enough.** We found that all these compression kernels are **memory-bound**. So we wrote **fused Triton kernels** that combine residual computation - compression - base updates into a single pass. This reduces memory traffic and keeps us within budget. (You may refer to `xfuser/compact/fastpath.py` in our anonymous repo.)
>
>  **Finally, we used some system-level tricks.** We integrated compression into the sequence parallel pipeline in a way that allows it to overlap with existing communication.
>
> **That’s why we’re able to get real speedups. It’s a combination of algorithm design, bottleneck analysis, and engineering efforts.**
>
> # Is CompactFusion Hard to Apply Because of These Optimizations? No.
>
> Although building a fast compressor was challenging, once we had a working implementation, it became easy to apply across different parallel strategies. Since CompactFusion operates purely at the communication layer, it is agnostic to the specific form of parallelism—all that’s needed is to wrap the communication calls. **This modularity allows the same compression logic to be reused with minimal effort across parallel methods.**
>
> # Measured Compression and Decompression Latency
>
> *Measured latency on L20, Flux:*
> | Latency of a single operation (ms)                | Compact-1bit | Compact-2bit | Compact-Lowrank | Compact-Sparse|
> |-------------------|--------------|--------------|------------------|----------------|
> | Compress FLUX     | 0.06         | 0.06         | 0.93             | 0.05           |
> | Decompress FLUX   | 0.02         | 0.02         | 0.04             | 0.04           |
> | Method | Quantization + Fused Kernel | Quantization + Fused Kernel | Subspace Iteration | N:M Sparsity
>
> ---
>
> We thank the reviewer for pointing out this important issue—compression latency was indeed one of the most challenging aspects of the system, and we’re glad to have the opportunity to elaborate on a part we invested significant effort into.

---

> ### Author Response · Authors · 2025-08-07
>
> We sincerely thank reviewer z8iV for the helpful feedback. It helped us improve the paper. We’ll include the discussion on compression latency in Appendix B, and we hope our response has addressed your concern.
>
>
> Best wishes!

---

### Official Review · Reviewer_ZQAB · 2025-07-02

**Clarity:** 2
**Significance:** 3
**Originality:** 3
**Rating:** 4
**Confidence:** 4

**Summary:**

Existing sequence parallelism-based methods enable reducing the sample latency of diffusion models by distributing computation across multiple devices. However, sequence parallelism introduces significant communication overhead, limiting its efficiency. This paper proposes CompactFusion, a framework that compresses the residuals of activations between adjacent denoising steps and introduces error feedback to mitigate cumulative global error. By transmitting compressed residuals instead of the raw activations, the communication cost is significantly reduced. Experiments on image and video generation demonstrate the effectiveness of the proposed method.

**Questions:**

- In Section 3.2, it is concluded that the error bound of residual compression is significantly smaller than that of naive compression, once we have $\delta>1-\frac{1-L^2}{L^2+1}$. Why is this a "mild condition"? Is there any quantitative result to support this statement for each compressor (1-bit quantization, 2-bit quantization, and low-rank approximation)?
- When generating a large number of samples (e.g., for FID evaluation) with several GPUs, will the proposed method be faster than naive sampling (i.e., each GPU generates a portion of samples without any parallel)?
- Could the authors provide a total execution time of compression, decompression and communication in one single sampling process?
- Typo: in line 141, ovear -> over; in line 45, paralleli -> parallelism.

**Ethical Concerns:**

["NO or VERY MINOR ethics concerns only"]

**Final Justification:**

I've read the authors’ response and the other reviewers’ comments, and I’m inclined to maintain my weak accept

**Limitations:**

Yes

**Paper Formatting Concerns:**

I don't find any major formatting issues in this paper.

**Quality:**

3

**Strengths And Weaknesses:**

Strengths:
- The idea of compressing the residuals of activations is supported by convincing evidence and is easy to follow.
- Compared with baseline methods, the proposed method exhibits superior performance on both speedup and sample quality.
- The proposed method can interact with existing sequence parallel methods easily, especially those communication-heavy methods.

Weaknesses:
- The writing of the preliminary part is overly simplified and could be improved. The proposed techniques, such as residual compression and error feedback, are applied to Sequence Parallel. However, the exact inference procedures are not introduced in the main content or the supplementary materials, making it challenging to understand the proposed method. Including more details of Sequence Parallel in the preliminary section, or including a more illustrative figure/algorithm to show the entire sampling process, could make the paper more readable.
- Though enjoying extremely small communication volume, the proposed Compact-Lowrank suffers from limited speedup brought by low-rank approximation. At the same time, its performance does not have an advantage compared to baseline methods, as well as Compact-1bit and Compact-2bit. Which scenarios is Compact-Lowrank suitable for?

---

> ### Author Rebuttal · Authors · 2025-07-30
>
> ## Thank you for your valuable comments and questions!
> We thank reviewer ZQAB for recognizing our contributions, including **strong evidence for residual activation compression, superior performance, and easy integration with existing parallel methods.**
> The reviewer also raised several constructive questions, which we address individually below.
>
> ---
>
> # Is Data Parallel (each GPU generates separately) Faster? It Depends.
> > When generating a large number of samples (e.g., for FID evaluation) with several GPUs, will the proposed method be faster than naive sampling (i.e., each GPU generates a portion of samples without any parallelism)?
>
> We thank the reviewer for raising this insightful question—we are happy to clarify this distinction in more detail.
>
> First, we note that the “naive sampling” described by the reviewer—where each GPU independently generates a subset of samples without communication—is **data parallel**.
>
> The key lies in how we define “faster.” There are two relevant notions:
>
> - **Throughput**: how quickly we can process a large batch of samples end-to-end.
>
> - **Latency**: how long it takes to generate a single sample from prompt to final output.
>
> **Interestingly, if our goal is throughput, which is likely the case in FID evaluation, then data parallelism (generates separately) will always be optimal.** This is not a limitation of CompactFusion; rather, it is a general property: any form of model-parallelism, including Tensor Parallel, Sequence Parallel, DistriFusion, PipeFusion, or CompactFusion—introduces coordination overhead and will typically have lower throughput than pure data parallelism.
>
> **However, if our goal is latency, then CompactFusion is faster.** CompactFusion enables multiple GPUs to collaboratively generate a single sample, reducing this sample's latency. While this introduces some overhead, it allows the sample to complete significantly faster than running it on a single GPU. In contrast, data parallelism cannot reduce latency per sample, since each image is generated independently by one GPU.
>
> **For user-facing applications, such as Sora and Midjourney, latency is often the main problem[1].** It directly determines how long a user waits after submitting a prompt before the final image/video is ready. This is exactly the problem that works such as DistriFusion and PipeFusion aim to tackle: reducing the per-sample latency with collaborative inference across multiple GPUs.
>
> Overall, this is a pretty interesting question that highlights a subtle but important distinction. We appreciate the reviewer for bringing it up.
>
> # Time Breakdown of CompactFusion
> > Could the authors provide a total execution time of compression, decompression and communication in one single sampling process?
>
> We thank the reviewer for the question and are happy to provide the requested breakdown.
>
> **Table: Component-wise Execution Time on CogVideoX (50 Steps, 6×L20)**
>
> | Component                 | Ring Attn (No Compression) | compact-1bit| compact-2bit| compact-lowrank|
> |--------------------------|-----------------------|--------|-------|----------|
> | Compression              | N/A                   | 0.61s  | 0.60s | 6.21s    |
> | Decompression            | N/A                   | 1.13s  | 1.15s | 1.92s    |
> | Comm Blocking Time       | 18.95s                | 1.05s  | 0.62s | 2.57s    |
> | Total Time           | 52.45s            | 35.74s | 35.81s| 43.58s   |
>
> **Table: Component-wise Execution Time on FLUX (28 Steps, 4×L20)**
>
> | Component             | Ring Attn (No Compression) | compact-1bit | compact-2bit | compact-lowrank |
> |----------------------|----------------------------|--------------|--------------|-----------------|
> | Compression          | N/A          | 0.20s        | 0.19s        | 2.96s           |
> | Decompression        | N/A        | 0.20s        | 0.21s        | 0.36s           |
> | Comm Blocking Time   | 4.07s        | 0.42s        | 0.48s        | 0.60s           |
> | Total Time           | 11.03s        | 7.52s        | 7.64s        | 10.66s          |
>
>
> Additional notes:
>
> 1. We reran the sampling process with a profiler to record detailed breakdowns for compression, decompression, and communication time as requested. As this is a separate run with profiling overhead, the total time shows slight variations compared to our original results. These variations do not affect the conclusions.
>
> 2. We observed that after enabling compression, the blocking time becomes very short, but it does not correlate directly with the compression ratio.
> We hypothesize that this is due to Ring Attn providing overlap, such that after compression is applied, communication can be fully hidden behind computation.
> As a result, the remaining blocking time is mostly due to startup and teardown overhead of the communication calls, rather than actual data transfer. This explains why the blocking time does not reflect the compression ratio, and instead fluctuates due to unrelated system factors.
>
> # Clarification on Bound Condition
> > In Section 3.2, it is concluded that the error bound of residual compression is significantly smaller than that of naive compression, once we have ...
>
> We thank the reviewer for raising this question regarding the condition in Section 3.2.
>
> This condition is a lower bound on compressor quality: unless the compressor is extremely poor, residual compression will yield a smaller error bound than naive compression.
>
> We call it mild because the tested $L$ is 0.14, which is relatively small; we only require the compressor’s $\delta$ to be greater than 0.056, a condition easy to satisfy for reasonable compressors. In practice, Compact‑1bit ($\delta \approx$ 0.58), Compact‑2bit ($\delta \approx$ 0.80), and Compact‑Lowrank ($\delta \approx$ 0.26) all satisfy it.
>
> Our experimental results on image quality strongly support this: even when using deliberately poor compressors, such as low‑rank with rank=8, only one‑quarter of the rank used in Compact‑Lowrank, residual compression is still better. In fact, based on all our experiments and various early trials, and to the best of our knowledge, we did not encounter a single case where naive compression performed better.
>
> We also note that the theoretical bound is derived under an idealized analytical setting to enable tractable analysis. Its role is to provide insight into why residual compression can outperform naive compression. Our main focus, however, is on validating this behavior under realistic settings, where our extensive experiments consistently confirm the theoretical prediction.
>
> # We Will Add More Preliminary on Parallel Strategies and Inference Procedures
> > The writing of the preliminary part is overly simplified and could be improved...
>
> **We thank the reviewer for highlighting this point. We agree that providing more preliminary about the inference procedure of sequence parallelism would help improve clarity and accessibility.**
>
> In the current version, we did not include a detailed explanation of sequence parallel, as our focus was primarily on the compression mechanism itself. Since **CompactFusion is designed to be parallel-strategy agnostic**, we aimed to keep the core method description decoupled from specific pipelines.
>
> That said, we agree that additional explanation would benefit readers—especially those less familiar with parallel inference.
>
> **Therefore, in the revision, we will include a dedicated section in the Appendix Preliminaries, providing:**
>
> - an overview of sequence parallelism and inference flow, including how activations are computed and exchanged;
>
> - illustrative figures and pseudocode showing how common parallel strategies (e.g., Ring Attention, Patch Parallel) work in practice;
>
> **In addition, we will add a separate section to clearly illustrate how CompactFusion integrates into these pipelines**, by wrapping communication points without modifying the execution logic.
>
> # Lowrank Compression Has Greater Potential in Constrained Networks
> > Though enjoying extremely small communication volume, the proposed Compact-Lowrank suffers from ... Which scenarios is Compact-Lowrank suitable for?
>
> Thank you for this excellent question. We believe Compact-Lowrank plays an important role in two key areas: edge-like low-bandwidth environments and stress-testing the robustness of residual compression. Below, we clarify its motivation.
>
> On PCIe interconnects, Compact-Lowrank shows limited speedup. This is expected for two reasons:
> (a) A 100× compression ratio is overkill for PCIe bandwidth;
> (b) Despite using subspace iteration for acceleration, low-rank compression is still slower. In this case, the overhead of compression outweighs the communication savings.
>
> However, in lower-bandwidth environments such as Ethernet, the benefit of extreme compression becomes more apparent. As shown in Figure 7.
>
> **We believe the lowrank method has greater potential in edge-level or constrained networks, where communication is even slower**. This would require adjusting the warmup strategy (need to use local warmup), which we describe in Appendix C: Warmup Implementation.
>
> **Compact-Lowrank also serves an important validation role to show the robustness of Residual Compression.** Despite its aggressive compression, it maintains better generation quality than DistriFusion (Table 1, 2), demonstrating the robustness of our method even under extreme ratios.
>
> Finally, part of our motivation was to test the boundary of residual compression: can we push to three-digit compression ratios while still outperforming the state-of-the-art overlap-based method in quality? We’re pleased to report that the answer(Table 1) is yes.
>
> ---
> **References**
>
> *[1] M. Li et al., “DistriFusion: Distributed Parallel Inference for High-Resolution Diffusion Models”.*

---

> ### Author Response · Authors · 2025-08-07
>
> We sincerely thank reviewer ZQAB for the thoughtful feedback. It helped us improve the clarity and completeness of our work.
>
> We hope our responses addressed your concerns, and as noted, we’ll include additional preliminaries in the Appendix to aid clarity and support readers unfamiliar with the parallel procedures.
>
> Thank you for your time. Warm regards.

---

### Note · Authors · 2025-08-13

We sincerely thank all reviewers and the AC for their time and thoughtful evaluation.

We are grateful for the recognition of CompactFusion’s key strengths:
- Clear improvements in latency and generation quality (QZGw, ZQAB).
- Tackling a fundamental challenge in balancing communication efficiency and perceptual quality (z8iV).
- Backed by empirical and theoretical support (ht4v, QZGw).
- Smooth integration into existing parallel systems (ht4v, ZQAB).

We also appreciate the constructive suggestions regarding preliminaries, compression latency, and temporal consistency, and will incorporate the corresponding improvements in the Appendix as promised.

Best regards,

The Authors

---

### Decision · Program_Chairs · 2025-09-17

**Decision:**

Accept (poster)

**Comment:**

This work is carefully reviewed and received consistent positive feedback (4445). All reviewers appreciate the observation/motivation on redundancy, the novelty of proposed solution and its effectiveness demonstrated on both image and video generation. Through the rebuttal, authors successfully addressed most concerned as acknowledged by reviewers. AC agrees that it does establish a new paradigm for parallel diffusion inference which is known to be quite important to push forward the efficient generation. Considering all the comments and discussions, a decision of acceptance is made. Authors are advised to include missing results raised by reviewers and necessary explanations for a strong and complete final version.